# Personality moderates associations between personal time and parental well-being
**Theresa Pauly** ✉

This study aimed to examine whether daily personal time—time spent free from external demands and available for self-directed activities—relates to better affective well-being and healthier cortisol patterns in midlife parents, and whether personality traits moderate these associations. A sample of 318 parents (Mage = 40.06 years, SD = 7.54; 45% men) with underage children (Mage of youngest child = 7.61 years, SD = 5.19) completed up to 8 consecutive days of daily diaries (mood, personal time, stress exposure) and up to 4 days of saliva sampling (4 times/day) for cortisol analysis. Multilevel modeling examined within-person links between personal time, positive and negative affect, and diurnal cortisol slopes, controlling for daily stress. Results showed that on days when they had an opportunity for time to themselves, parents experienced higher positive affect, lower negative affect, and steeper cortisol slopes, indicating better stress recovery. The reduction in negative affect with personal time was stronger for parents high in neuroticism and openness, and high neuroticism was also linked with a stronger association between personal time and cortisol slopes. Findings underscore the potential restorative value of daily time to oneself for midlife parents, particularly those high in neuroticism and openness. In the context of the high demands of parenting, personal time may serve as a valuable resource for emotional renewal, solitude, self-care, self-connection, and recovery from daily parenting stress.

Many adults report feeling short on personal or leisure time[1,2]. A subjective feeling of lack of personal time has been associated with diminished well-being and lower quality of life[3,4]. Time strains are particularly pronounced for midlife parents, who are often juggling multiple roles at once—managing family responsibilities, caring for aging parents, and working in high-demand career stages[5,6]. This study explores the role of personal time for the daily emotional and physiological well-being of parents of underage children, and examines whether these associations are moderated by personality.

Personal time refers to time that an individual spends in self-directed, restorative or voluntary activities, free from obligatory work, caregiving, or household duties[7,8]. Personal time can but does not necessarily need to happen when alone, defined as the physical absence of other people, or in solitude, defined as the lack of social interaction[9]. Early research on personal time emphasized privacy—the selective control of access to the self[10]—as a way individuals control interactions with others, uphold personal boundaries to preserve self-identity, and regulate well-being. Pedersen[11] identified five core functions of privacy that may be facilitated by personal time: autonomy (independence, freedom from social pressure), confiding

(intimacy, expressing true emotions), rejuvenation (taking refuge, recover), contemplation (self-reflection), and creativity (problem-solving, idea generation). The Social Affiliation Model[12] offers a homeostatic view of time to oneself and social interaction. Much like hunger drives food intake, humans are thought to monitor and regulate social engagement, seeking to maintain an optimal range. When this balance is disrupted—either by too much time to oneself or too much social contact—people are motivated to reestablish equilibrium. Similarly, the Communicate Bond Belong Theory[13] suggests that social engagement is governed by both the desire to belong and the need to conserve energy. Individuals are thought to have a limited amount of social energy, and when it is depleted, they are more likely to seek time to oneself. In the life-balance literature, research indicates that a three-dimensional model—including work time, social time, and personal time—better predicts health outcomes than a two-dimensional model based only on work and social time[14].

Research has linked engaging in leisure activities or activities that are driven by intrinsic motivation with enhanced recovery, defined as the process to replenish resources like energy, attention, and mood when not exposed to further demands[15,16]. Recent work on the well-being benefits of

Department of Gerontology, Simon Fraser University, Vancouver, BC, Canada. ✉e-mail: theresa_pauly@sfu.ca

personal time has focused on *positive solitude*. Palgi et al.[17] describe positive solitude as situations where individuals intentionally choose to spend time alone, such as reading, going for a walk, or working on their computer, or find ways to enrich their solitude by engaging in personally meaningful or enjoyable activities like listening to music. Similarly, Ost Mor et al.[18] conceptualize positive solitude as "the choice to dedicate time to a meaningful, enjoyable activity or experience conducted by oneself" (p. 15), emphasizing that such experiences may be spiritual, recreational, or functional in nature. Individuals who report higher positive solitude, report lower levels of depressive symptoms[19], higher levels of mindfulness[20] and flourishing[21], and better health and well-being[17]. Time to oneself is often inherently enjoyable, suggesting its restorative effects may partly operate through positive emotions[8]. Experience sampling studies that track people's emotions throughout their daily lives have found that individuals tend to report greater levels of low-arousal positive emotions, such as calmness, contentment, and relaxation, when in solitude or when engaging in leisure activities by themselves[22–24].

Conversely, the subjective perception of not enough time for oneself has been associated with increased levels of negative affect, stress, and depressive symptoms[25] and lower levels of life and family satisfaction[5]. Without enough personal time, individuals may struggle to decompress, reflect, or disengage from daily demands. Personal time might contribute to well-being through a number of mechanisms[26]: (a) Competence: Time to oneself fosters skill development, self-efficacy, and the pursuit of meaningful leisure activities; (b) Autonomy: Time to oneself provides a sense of choice, freedom from social pressure, and relief from external responsibilities, enabling deeper self-connection; (c) Self-growth: Time to oneself supports self-reflection, the development of coping strategies, and spiritual or existential exploration; and (d) Self-care: Many use time to oneself to recharge and attend to physical, mental, and emotional needs. One demographic for which personal time might be particularly scarce is for adults in midlife who are raising children.

The transition to parenthood—especially during a child's first year—results in profound shifts in daily routines, personal identity, and time use[27]. Parents consistently report an overwhelming reorganization of their lives as caregiving responsibilities become all-consuming[27,28]. Daily care for young children often demands near-constant attention, leaving little time for personal needs, rest, or recuperation[28]. These changes are not only logistical but deeply emotional and psychological; parents often struggle to recalibrate their roles and routines, and to adjust to the loss of their former life and activities[27]. Parenting continues to impose substantial and often intensifying demands throughout childhood and adolescence. Midlife parents frequently manage peak career responsibilities while coordinating children's school schedules, extracurricular activities, and evolving socioemotional needs, leaving little discretionary time for personal recovery[29]. Thus, not surprisingly, parental stress is robustly linked to reduced well-being[30] and can lead to parental burnout, a condition characterized by chronic exhaustion related to parenting, emotional distancing from one's children, and a sense of being an ineffective parent[31].

One of the most pronounced consequences of parenthood is the reduction in leisure activities and personal time. Multiple studies show that both mothers and fathers experience a substantial decline in recreational and restorative activities[5,6,32,33]. However, mothers tend to bear a disproportionate burden of this shift. Compared to fathers, they are twice as likely to report that they do not have enough time for themselves[5], exercise less frequently[33], and report more difficulty mentally disengaging from caregiving demands and other household responsibilities even when physically alone[34,35].

Consequently, having enough time to oneself might have important implications for well-being in parents. Parental stress often involves feelings of being overwhelmed, anxious, and depleted—emotions that time to oneself can help down-regulate[22,35]. Personal time also allows for energy restoration and mental reset[13,22]. Even taking brief amounts of time for oneself—such as napping, resting, or listening to calming music—might provide relief from daily demands. These moments of personal space might help replenish depleted resources and restore the energy required for continued caregiving[35].

Previous research has primarily examined the links between personal time or leisure activities with parents' health and well-being using either cross-sectional methods or long-term longitudinal designs that span multiple years e.g.,[5,33]. However, these approaches fall short in capturing the dynamic, short-term, and context-dependent nature of time to oneself—especially its potential function as a source of restoration and emotional regulation.

The current study addresses this limitation by employing an experience sampling design, which offers a more fine-grained, real-time window into how variations in daily personal time relate to fluctuations in psychological and physiological well-being. By tracking participants' experiences across multiple days, this method captures within-person variability and reveals how personal time may function as a coping resource on a day-to-day basis, rather than assuming stable, trait-like associations[36].

Furthermore, prior research has rarely investigated *how* time to oneself might protect from long-term ramifications of parenting stress on physical health. The present study begins to fill this gap by examining daily cortisol patterns—a key biomarker of stress and recovery[37]. Cortisol marks activity of the hypothalamic-pituitary-adrenal (HPA) axis and follows a diurnal rhythm, peaking shortly after waking and declining steadily throughout the day[38]. On days with acute stress or heightened negative emotions, this rhythm flattens, signaling reduced physiological recovery[39,40]. Overall, flatter diurnal cortisol slopes have been linked to a wide range of adverse mental and physical health outcomes, including increased risk for depression, impaired immune functioning, obesity, cancer, and cardiovascular disease[41,42]. Even single- or few-day measures of flattened diurnal cortisol slopes are linked to negative health effects, suggesting that sustained reductions in physiological recovery may gradually lead to maladaptive HPA axis regulation[41]. HPA axis dysregulation, in turn, can result in abnormal total cortisol output, either excessive (hypersecretion) or diminished (hyposecretion)[43]. While leisure activities such as music listening have been shown to reduce cortisol in everyday life[44], the potential stress-buffering effects of personal time during daily life remain understudied. This study examined whether personal time is linked with lower daily negative affect and steeper cortisol slopes in parents of underage children. However, the nature of these associations may vary from one parent to another.

Personality might play a role in shaping responses to the stresses of parenthood. Longitudinal and cross-sectional studies alike show that individuals high in neuroticism—characterized by heightened emotional instability, vulnerability to stress, and anxiety—are more likely to experience elevated parental stress and burnout[45–47]. Conversely, extraversion—characterized by energy, positive emotions, sociability, and the tendency to seek stimulation and the company of others—is generally protective with extraverted parents report lower levels of parenting stress[47]. Conscientiousness—characterized by self-discipline, carefulness, thoroughness, organization, and a desire to achieve goals—has been related both reduced risk[46] as well as increased risk (particularly for the trait of meticulousness[40]) of parental burnout.

Individual differences might thus also play a role in who seeks and benefits from personal time when faced with parental stress. Traits such as sensory processing sensitivity and affinity for aloneness have been linked to a greater tendency to seek time to oneself and benefit from it emotionally[48,49]. Among the Big Five personality traits, introversion—characterized by lower sociability and assertiveness, and greater tendencies toward reserved, reflective, and solitary behavior—would intuitively seem most closely linked to the experience of time to oneself[50]. However, several studies have found no significant relationship between introversion and enjoyment of solitude, self-determined motivation for solitude, or preference for solitude[48,49,51]. More recent research with a representative U.S. sample (N = 501) has shown that personality traits might relate differentially to solitude functions[52], with extraversion and neuroticism showing the strongest associations. Specifically, the authors found that individuals high in neuroticism viewed solitude as important for emotion regulation and escape, while those high in

extraversion placed less value on solitude for relaxation or avoiding unpleasant interactions. Openness, on the other hand, was linked to using solitude for creativity and self-discovery[52]. Consequently, another aim of this study is to examine whether daily associations between time to oneself and negative affect as well as cortisol slopes are moderated by personality traits.

The present study investigates how personal time relates to the daily emotional and physiological well-being of midlife parents. Drawing on experience sampling data from a subsample ($N = 318$) of the Midlife Development in the United States (MIDUS) Study, collected between 2011 and 2014, participants reported their daily experiences across eight consecutive days, with salivary cortisol samples collected across 4 days. This study focused on diurnal cortisol slopes as an index of daily stress physiology. Unlike other cortisol measures (e.g., area-under-the-curve, awakening response), a flattened slope specifically reflects reduced recovery, indicating impaired down-regulation of cortisol by evening[42]. Daily stress exposure was controlled for to account for its potential confounding influence on affect and cortisol[37,40], as busier, more stressful days may limit opportunities for personal time. This allows the unique association of personal time with daily well-being to be examined, holding daily stress exposure constant. Guided by recent work highlighting the role of individual differences in the function and experience of time to oneself[52], the study further examined how the associations between personal time and daily well-being may be moderated by personality.

It was hypothesized that (H1) on days when individuals have an opportunity for time to themselves, they report higher positive affect and lower negative affect and exhibit steeper diurnal cortisol slopes compared to days without time to themselves. Furthermore, (H2) these associations would vary depending on personality traits. Directed hypotheses regarding personality traits were not specified because the existing literature on personality differences in the experience of solitude is inconsistent (e.g., with respect to introversion[48,49,51]), and even less is known about how personality relates specifically to the benefits of personal time. Hypotheses were not preregistered.

## Methods
### Procedures and participants
Data for the current study are from the Midlife Development in the United States Survey (MIDUS; http://midus.wisc.edu), which began in 1994 and has collected data from the same participants across three waves since (MIDUS1: 1994–1995, $N = 7108$ adults aged 25–74; MIDUS2: 2004–2006, $N = 5555$; MIDUS3: 2013–2015, $N = 3683$). Between 2011 and 2014, an additional sample (Refresher Cohort) of 3577 adults aged 25–74 was recruited to replenish the number of middle-aged adults in the original MIDUS cohort. Participants were recruited through random digit dialing. The MIDUS Refresher survey used the same assessments as the original study, where participants first completed 30 min baseline phone interviews followed by self-administered questionnaires via mail. A random subsample of participants ($N = 782$) was selected to take part in a Daily Diary project. On 8 consecutive evenings, participants completed phone surveys about their experiences over the past day. These daily surveys covered various aspects of their daily lives, including stressors, emotions, physical symptoms, and social interactions. The sociodemographic characteristics of participants in the Daily Diary study closely resembled those of the broader Refresher survey sample[53]. Participants provided informed consent and MIDUS data collection is reviewed and approved by the Education and Social/Behavioral Sciences and the Health Sciences IRBs at the University of Wisconsin-Madison. Secondary data analysis for this study was approved by the Research Ethics Board at Simon Fraser University. Participants were compensated $25 for completing the Daily Diary protocol.

The current study uses data from the Refresher Daily Diary subproject only, as daily time to oneself was not measured in any of the three other MIDUS surveys. Out of the 782 participants, 319 individuals had at least one underage child living in their household (biological child, adopted child, stepchild, or child of partner). One person was missing information on personality, resulting in a sample size of $N = 318$ participants who provided

a total of $n = 2299$ surveys. Participants had an average age of 40.06 years ($SD = 7.54$), 55% were women and 45% were men, and the average annual household income was $97,435.71 ($SD = $65,183.49$). They lived with an average of 2 underage children ($M = 2.02$, $SD = 1.15$) and the youngest child was 7.61 years old, on average ($SD = 5.19$). The majority of the sample identified as white (86.5%), 5.0% as Black, 1.3% as native American, 0.9% as Asian, and 6.3% as other. Participants rated their subjective health as fairly good ($M = 7.52$, $SD = 1.48$) on a 0–10 scale, where 0 indicated "the worst possible health" and 10 indicated "the best possible health." Participants reported experiencing any stressor on approximately half of the days ($M = 0.47$, $SD = 0.28$). The most commonly reported stressor was avoiding a disagreement ($M = 1.37$ out of 8, $SD = 1.33$), followed by an argument or disagreement ($M = 1.03$, $SD = 1.19$), a stressor at work or school ($M = 0.93$, $SD = 1.13$), a stressful event at home ($M = 0.65$, $SD = 1.02$), a stressful event that happened to a close other ($M = 0.30$, $SD = 0.62$), and discrimination ($M = 0.02$, $SD = 0.12$).

Wilcoxon rank sum tests showed that this sample of parents was significantly younger ($W = 119'315$, $p < 0.001$), had higher average household income ($W = 54'798$, $p < 0.001$), better self-rated health ($W = 65'722$, $p = 0.013$), less time to themselves ($W = 93'828$, $p < 0.001$), lower average negative affect ($W = 67'304$, $p = 0.035$), and steeper cortisol slopes ($W = 63'112$, $p < 0.001$) than the remainder of the Refresher Daily Diary sample. Samples did not significantly differ by gender ($\chi^2$-test; $\chi^2 = 0.19$, $p = .666$). Participants completed 7.23 out of 8 scheduled surveys ($SD = 1.66$) and provided 3.08 ($SD = 1.02$) out of 4 days of valid cortisol data, on average.

### Measures
**Personal Time.** As part of the 8 daily surveys, participants were asked: "Since this time yesterday, did you have the opportunity to take time for yourself?". Answer options were "Yes" (1) or "No" (0).

**Daily affect.** Participants were also asked to report their mood using 27 items related to positive and negative affect, with the prompt: "How much of the time today did you feel …?" Responses were rated on a scale from 0 (not at all) to 4 (all the time). Positive affect items included: in good spirits, cheerful, extremely happy, calm and peaceful, satisfied, full of life, close to others, like you belong, enthusiastic, attentive, proud, active, and confident. Negative affect items included: restless or fidgety, nervous, worthless, so sad nothing could cheer you up, everything was an effort, hopeless, lonely, afraid, jittery, irritable, ashamed, upset, angry, and frustrated. Items were drawn from the Nonspecific Psychological Distress Scale[54] and a modified version of the Positive and Negative Affect Schedule[55]. Daily positive and negative affect were calculated by averaging the scores across all respective items, with higher scores indicating higher levels of positive or negative affect (possible range: 0–4). Reliability indices of repeated measures as described by Cranford et al.[56] were high at both the between-person (positive affect: $R_{KF} = 0.96$; negative affect: $R_{KF} = 0.86$) and within-person level (positive affect: $R_C = 0.89$; negative affect: $R_C = 0.84$).

**Diurnal cortisol slope.** Participants collected four saliva samples on days two through five of their 8-day daily life assessments at the following times: immediately upon waking, 30 min post-waking, before lunch, and before bedtime. Collection times were logged and verified through daily diary interviews. Saliva samples were analyzed for cortisol with a commercially available luminescence immunoassay (IBL, Hamburg, Germany) at the Biological Psychology Laboratory at the Technical University of Dresden. Inter-assay and intra-assay coefficients of variance were below 5%. For more details about cortisol collection and processing, see Almeida et al.[57] and Dmitrieva et al.[58]. Cortisol data were excluded if participants demonstrated non-compliance or out-of-range cortisol values. In line with previous MIDUS studies[58], any sample with cortisol ≥60 nmol/L was treated as missing, and days were excluded if participants awoke before 4 a.m. or after 11 a.m., were awake for less than

12 h or more than 20 h, if the first waking sample was delayed by more than 15 min after waking, or if cortisol levels increased by more than 10 nmol/L at bedtime compared to post-waking levels.

Cortisol data were available from 269 individuals. Fourteen individuals were excluded because they did not provide at least one valid day of cortisol data (i.e., a waking and a bedtime sample). Thus, the final analytic sample consisted of 788 valid cortisol days, completed by 255 participants. Wilcoxon and $\chi^2$ tests showed that those excluded due to non-compliance did not significantly differ from included participants in terms of gender ($\chi^2 = 0.41$, $p = 0.524$), ethnicity ($\chi^2 = 1.24$, $p = 0.266$), age ($W = 2'324$, $p = 0.562$), income ($W = 2'428$, $p = 0.325$), subjective health ($W = 1'811$, $p = 0.342$), average personal time ($W = 1'589$, $p = 0.094$), or negative affect ($W = 1'986$, $p = 0.673$), but they did report significantly higher positive affect ($W = 1'422$, $p = 0.036$). On average, participants provided 3.08 days (SD = 1.02) of cortisol data. Cortisol values were log10-transformed. For each day, the diurnal slope was calculated by subtracting the morning from the evening cortisol value, dividing by the number of hours between the two samples[59]. Dividing by time elapsed between samples accounts for differences in total time awake—reflecting the duration over which cortisol can decline—and is the recommended method for estimating diurnal cortisol slopes[42]. The wake–evening cortisol slope is the most commonly used marker of diurnal cortisol decline and less vulnerable to distortion from daytime outliers, as compared to regression-based estimation methods that use all daily samples[41].

**Personality**. Participants rated how well 26 self-descriptive adjectives described them on a scale from 1 ("a lot") to 4 ("not at all"). These adjectives assessed the Big 5 personality traits: neuroticism (4 items), extraversion (5 items), openness to experience (7 items), conscientiousness (5 items), and agreeableness (5 items). For information on scale construction see Lachman and Weaver[60]. Personality trait scores were calculated as the mean of the relevant items, with some items reverse-coded so that higher scores indicated greater expression of each trait. Scores were computed for participants with valid responses on at least half of the items for a given trait. All scales demonstrated sufficient reliability (Cronbach's $\alpha$ 0.69 to 0.79).

**Covariates**. In the daily surveys, participants reported whether they had encountered a stressful event in any of the following domains of life that day: argument/disagreement, stressful event at work/school, stressful event at home, stressful event that happened to a friend, other stressful event. To control for stress exposure, a binary variable was created that indicated whether participants had encountered any stressor that day (0 = no stressful event, 1 = at least one stressful event) and the average amount of stress exposure throughout the study period (person-mean stress exposure). In the mail-out surveys, participants reported their age, gender, household income, the number of children aged <18 years living in the household, the age of the youngest child, and their self-rated health (0 = worst possible health, 10 = best possible health). Additionally, participants reported whether they had taken any medications known to influence cortisol on the days they provided saliva (including steroid nasal sprays, steroid inhalers, oral steroids, cortisone creams, corticosteroid injections, antidepressants or anti-anxiety medications, or other hormonal medications). A binary variable was created with 0 = no hormonal medication, 1 = at least 1 hormonal medication.

**Data Analysis**
Two-level multilevel models with days (level 1) being nested within participants (level 2) were computed using the lme4 package[61] in R Studio 2024.12.1 (running R 4.4.2[62];). To disentangle between and within-person effects, personal time was person-mean centered prior to analysis. Person-mean personal time (aggregated over all 8 days) was entered on the day level to examine between-person effects. Continuous between-person (level 2) variables were centered on the sample mean. Models control for temporal changes in the outcome (study day; first day coded as 0) and include the

random slope of daily personal time. Standardized regression estimates and pseudo-R2 were calculated for effect sizes. All analysis code can be accessed on the Open Science Framework (https://osf.io/an5ts/).

A first set of models predicted daily affect and diurnal cortisol slopes by study day, daily personal time, and stressor exposure at level 1 and by person-mean personal time, person-mean stressor exposure, and the five personality traits at level 2 (main effects only model; H1). In a second step, all five interaction terms between daily personal time and personality traits were entered simultaneously into the models at level 2 (moderation model; H2). Significant interactions were probed by calculating simple slopes at one standard deviation below ($M - 1$ SD) and above ($M + 1$ SD) the mean of the moderator. Sensitivity analyses tested whether findings remained consistent when including all the following additional covariates into the models: age, income, number of children, age of youngest child (all models); medication intake, subjective health (models for cortisol). Model assumptions and fit, including normality of residuals, homogeneity of variance, multicollinearity, and the presence of influential observations, were evaluated using the *performance* package in R. No assumption violations were detected.

Simulations using the simr package were conducted to calculate statistical power for this study[63]. Based on 318 participants providing an average of 7 out of 8 daily surveys and using an alpha-error rate of 5%, this study was able to detect small within-person effects of $r = 0.10$ with about 94% power and moderate cross-level interactions of $r = 0.30$ with about 90% power. For diurnal cortisol data, a sample of $N = 255$ participants provided 71% power to detect small within-person effects and 61% power to detect moderate cross-level interactions.

## Results
### Descriptive Results
Table 1 presents descriptive statistics and within- and between-person correlations of study variables. Participants reported having had the opportunity to take personal time on 79% of days (SD = 0.25), averaging about 2 days out of every 8 days without personal time. This was not significantly different between men ($M = 0.79$, SD = 0.24) and women ($M = 0.80$, SD = 0.26; $t = 0.33$, $p = 0.738$). Average positive affect was moderate to high ($M = 2.43$, SD = 0.72) while negative affect was close to the lower end of the scale ($M = 0.23$, SD = 0.21). The average diurnal cortisol slope was $-0.61$ (SD = 0.22).

Regarding bivariate between-person associations, older age ($r = 0.18$; [0.07, 0.28]), having older children ($r(316) = 0.16$; [0.05, 0.26]), and experiencing fewer stressors ($r(316) = -0.11$; [$-0.22$, $-0.01$]) were significantly associated with more personal time. Taking medication ($r(316) = 0.14$; [0.03, 0.24]), worse self-rated health ($r(316) = -0.23$; [$-0.34$, $-0.13$]), and higher stressor exposure ($r(316) = 0.50$; [0.41, 0.58]) were associated with higher average levels of negative affect. Better self-rated health ($r(316) = 0.26$; [0.15, 0.36]), and lower stressor exposure ($r(316) = -0.27$; [$-0.37$, -0.17]) were associated with higher average levels of positive affect. Participants who were older ($r(253) = 0.16$; [0.03, 0.27]) and those who had older children ($r(253) = 0.18$; [0.06, 0.29]) showed less steep cortisol slopes.

We ran unconditional models to identify variance composition at the day and person level. All four daily assessed parameters showed significant variation on a day-to-day level (personal time: 75%; positive affect: 25%; negative affect: 61%; cortisol slope: 73%; daily stressors: 84%). Bivariate within-person correlations showed that on days when participants reported personal time, they reported higher levels of positive affect ($r(1980) = 0.11$; [0.06, 0.15]), lower levels of negative affect ($r(1980) = -0.09$; [$-0.13$, $-0.05$]), and steeper cortisol slopes ($r(532) = -0.11$; [$-0.19$, $-0.03$]). On days when negative affect was increased, participants showed less steep cortisol slopes ($r(532) = 0.09$; [0.01, 0.17]). On days, when participants reported a stressor, they reported lower levels of positive affect ($r(1980) = -0.18$; [$-0.22$, $-0.14$]), higher levels of negative affect ($r =(1980)$ 0.35; [0.31, 0.38]), and less personal time ($r(1980) = -0.06$; [$-0.10$, $-0.01$]).

**Table 1 | Descriptive statistics and correlations among study variables (N = 318)**

| Variable | M | SD | 1 | 2 | 3 | 4 | 5 | 6 | 7 | 8 | 9 | 10 | 11 | 12 |
|---|---|---|---|---|---|---|---|---|---|---|---|---|---|---|
| 1. Age | 40.06 | 7.54 | | | | | | | | | | | | |
| 2. Gender (1 = man) | 0.45 | 0.50 | 0.04 | | | | | | | | | | | |
| 3. Household income | 97,434.71 | 65,183.49 | 0.20** | 0.21** | | | | | | | | | | |
| 4. # children in household | 2.02 | 1.15 | −0.15** | −0.01 | 0.01 | | | | | | | | | |
| 5. Age of youngest child | 7.61 | 5.19 | 0.64** | −0.13* | −0.02 | −0.39** | | | | | | | | |
| 6. Taking any medicine (1 = Yes) | 0.32 | 0.47 | 0.15** | −0.11 | −0.00 | −0.10 | −0.10 | | | | | | | |
| 7. Self-rated health (0–10) | 7.52 | 1.48 | −0.05 | −0.04 | 0.10 | −0.00 | 0.12* | −0.15** | | | | | | |
| 8. Stressor exposure | 0.47 | 0.28 | −0.05 | −0.10 | −0.01 | 0.13* | −0.10 | 0.02 | −0.05 | | −0.06* | −0.18** | 0.35** | 0.07 |
| 9. Personal time (0–1) | 0.79 | 0.25 | 0.18** | −0.02 | 0.00 | −0.03 | 0.16** | −0.06 | 0.05 | −0.11* | | 0.11** | −0.09** | −0.011* |
| 10. Positive affect (0–4) | 2.43 | 0.72 | −0.06 | −0.02 | 0.03 | −0.02 | −0.01 | −0.08 | 0.26** | −0.27** | 0.09 | | −0.49** | |
| 11. Negative affect (0–4) | 0.23 | 0.21 | 0.03 | −0.07 | −0.02 | 0.05 | −0.05 | 0.14* | −0.23** | 0.50** | −0.09 | −0.46** | | 0.09* |
| 12. Diurnal cortisol slope | −0.61 | 0.22 | 0.16* | −0.04 | 0.01 | −0.01 | 0.18** | 0.06 | −0.09 | −0.05 | 0.07 | 0.05 | −0.02 | |

Note. Means and SD are person-weighted averages for personal time, negative affect, and diurnal cortisol slope. Between-person correlations are below and within-person correlations are above the diagonal. * p < 0.05, ** p < 0.01.

## Main results

### H1: Personal time as a predictor of affect and cortisol.

Multi-level models predicting affect and salivary cortisol by personal time and personality, controlling for stressor exposure and study day can be found in Table 2. As hypothesized, parents were more likely to report higher levels of positive affect ($\beta = 0.05$, SE = 0.01, 95% CI = [0.02, 0.07], $p < 0.001$), lower levels of negative affect ($\beta = −0.05$, SE = 0.02, 95% CI = [−0.08, −0.01], $p = 0.005$), and steeper diurnal cortisol slopes ($\beta = −0.10$, SE = 0.03, 95% CI = [−0.17, −0.03], $p = 0.005$) on days on which they had the opportunity for time to themselves (see Fig. 1). Daily stressor exposure was associated with decreased levels of positive affect ($\beta = −0.11$, SE = 0.01, 95% CI = [−0.14, −0.09], $p < 0.001$) and increased levels of negative affect ($\beta = 0.27$, SE = 0.02, 95% CI = [0.24, 0.31], $p < 0.001$) but not with cortisol slopes ($\beta = 0.07$, SE = 0.04, 95% CI = [0.00, 0.14], $p = 0.061$). Person-mean personal time and person-mean stressor exposure were not significantly related to cortisol slopes, but participants with higher person-mean stressor exposure reported decreased levels of positive affect ($\beta = −0.14$, SE = 0.04, 95% CI = [−0.22, −0.05], $p = 0.001$) and increased levels of negative affect, on average ($\beta = 0.16$, SE = 0.03, 95% CI = [0.10, 0.23], $p < 0.001$).

With respect to personality, higher levels of neuroticism ($\beta = 0.21$, SE = 0.03, 95% CI = [0.15, 0.28], $p < 0.001$) and openness ($\beta = 0.09$, SE = 0.04, 95% CI = [0.02, 0.16], $p = 0.010$) and lower levels of extraversion ($\beta = −0.09$, SE = 0.04, 95% CI = [−0.17, −0.02], $p = 0.016$) were linked to higher average levels of negative affect. Lower levels of neuroticism ($\beta = −0.11$, SE = 0.04, 95% CI = [−0.20, −0.02], $p = 0.012$), higher levels of extraversion ($\beta = 0.31$, SE = 0.05, 95% CI = [0.20, 0.42], $p < 0.001$), and higher levels of conscientiousness ($\beta = 0.17$, SE = 0.05, 95% CI = [0.08, 0.25], $p < 0.001$) were linked to higher average levels of positive affect. Parents with lower levels of conscientiousness ($\beta = 0.11$, SE = 0.05, 95% CI = [0.02, 0.20], $p = 0.023$), lower levels of agreeableness ($\beta = 0.14$, SE = 0.05, 95% CI = [0.04, 0.25], $p = 0.009$), and higher levels of openness ($\beta = −0.12$, SE = 0.05, 95% CI = [−0.23, −0.02], $p = 0.021$) showed steeper cortisol slopes, on average.

Explained variance in positive affect was 22.6% for fixed effects and 76.2% for fixed and random effects. Explained variance in negative affect was 24.3% for fixed effects and 47.2% for fixed and random effects. Explained variance in diurnal cortisol slopes was 5.4% for fixed effects and 36.9% for fixed and random effects.

To explore potential time-ordered relationships, time-lagged effects from one day to the next were tested. Personal time on the previous day was not significantly associated with next-day positive affect ($\beta = 0.01$, SE = 0.01, 95% CI = [−0.01, 0.03], $p = 0.485$), negative affect ($\beta = 0.00$, SE = 0.02, 95% CI = [−0.03, 0.03], $p = 0.905$), or cortisol slope ($\beta = 0.01$, SE = 0.03, 95% CI = [−0.05, 0.08], $p = 0.700$) after accounting for current-day personal time and both current and prior day stress exposure.

### H2: Personality as a moderator of associations of personal time with daily well-being.

Next, interactions between personal time and personality in predicting affect and salivary cortisol were introduced into the multi-level models to test for moderating effects. The tables containing all parameters are included in the online supplement. Daily associations between opportunity for personal time and positive affect were not significantly moderated by personality ($\beta$'s = −0.00 to 0.02, SE's = 0.01, 95% CI's = [−0.03, 0.04], $p$'s = 0.175 to 0.884; see Supplementary Table 1). The within-person association between opportunity for personal time and negative affect was moderated by neuroticism ($\beta = -0.05$, SE = 0.02, 95% CI = [−0.09, −0.01], $p = 0.006$) and openness ($\beta = −0.04$, SE = 0.02, 95% CI = [−0.08, 0.00], $p = 0.035$). As can be seen in Fig. 2, individuals high in neuroticism ($\beta = −0.09$, SE = 0.02, 95% CI = [−0.13, −0.05], $p < 0.001$) and in openness ($\beta = −0.09$, SE = 0.02, 95% CI = [−0.14, 0.04], $p < 0.001$) experienced a significant decrease in negative affect on days when they reported having had time to themselves. This was not true for individuals low in neuroticism ($\beta = 0.00$, SE = 0.02, 95% CI = [−0.05, 0.05], $p = 0.937$) or openness ($\beta = −0.02$, $SE = 0.02$, 95% CI = [−0.06, 0.02], $p = 0.408$). When all personality traits were entered into the same model,

**Table 2 | Multi-level models evaluating personal time and personality as a predictor of daily positive/negative affect (n = 2299 observations of N = 318 individuals) and salivary cortisol (n = 788 observations of N = 255 individuals)**

|  | Positive Affect | | | Negative Affect | | | Diurnal Cortisol Slope | | |
|---|---|---|---|---|---|---|---|---|---|
| *Predictors* | β | SE | p | β | SE | p | β | SE | p |
| (Intercept) | 0.02 | 0.04 | 0.715 | 0.00 | 0.03 | <0.001 | −0.00 | 0.05 | <0.001 |
| *Level-1 Fixed Effects* | | | | | | | | | |
| Personal time | 0.05 | 0.01 | <0.001 | −0.05 | 0.02 | 0.005 | −0.10 | 0.03 | 0.005 |
| Stressor exposure | -0.11 | 0.01 | <0.001 | 0.27 | 0.02 | <0.001 | 0.07 | 0.04 | 0.061 |
| Day in Study (0 = first day) | -0.05 | 0.01 | <0.001 | −0.07 | 0.02 | <0.001 | 0.05 | 0.03 | 0.120 |
| *Level-2 Fixed Effects* | | | | | | | | | |
| Person mean personal time | 0.03 | 0.04 | 0.447 | −0.03 | 0.03 | 0.360 | 0.07 | 0.05 | 0.132 |
| Person mean stressor exposure | −0.14 | 0.04 | 0.001 | 0.16 | 0.03 | <0.001 | −0.04 | 0.05 | 0.372 |
| Neuroticism | −0.11 | 0.04 | 0.012 | 0.21 | 0.03 | <0.001 | 0.02 | 0.05 | 0.739 |
| Extraversion | 0.31 | 0.05 | <0.001 | −0.09 | 0.04 | 0.016 | −0.01 | 0.06 | 0.885 |
| Openness | −0.02 | 0.05 | 0.662 | 0.09 | 0.04 | 0.010 | −0.12 | 0.05 | 0.021 |
| Conscientiousness | 0.17 | 0.04 | <0.001 | −0.06 | 0.03 | 0.055 | 0.11 | 0.05 | 0.023 |
| Agreeableness | −0.01 | 0.05 | 0.793 | 0.07 | 0.04 | 0.058 | 0.14 | 0.05 | 0.009 |
| *Random Effects* | | | | | | | | | |
| Level-1 residual variance | 0.16 | | | 0.05 | | | 0.05 | | |
| Random intercept variance | 0.36 | | | 0.02 | | | 0.02 | | |
| Random slope variance (personal time) | 0.01 | | | 0.01 | | | 0.02 | | |
| Correlation random intercept & slope | 0.00 | | | −0.54 | | | 0.22 | | |
| Marginal R²/Conditional R² | 0.226/0.762 | | | 0.243/0.472 | | | 0.054/0.369 | | |

All continuous predictors were grand-mean centered. Personal time was within-person-centered. Marginal and conditional $R^2$ values represent the variance explained by fixed effects alone and by the full model, respectively.

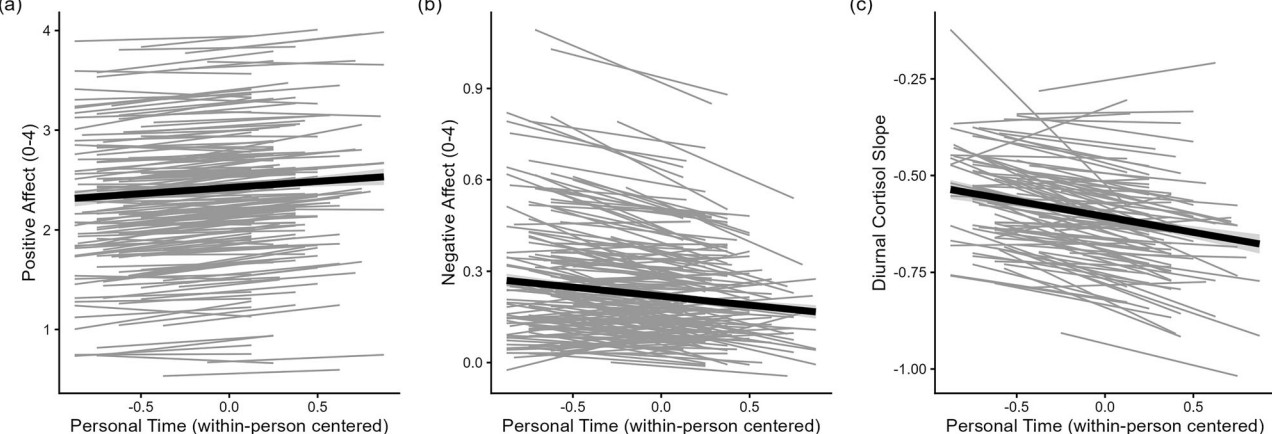

**Fig. 1 | Associations of personal time with daily affective well-being and salivary cortisol.** Parents showed a greater increase in positive affect (**a**), a greater decrease in negative affect (**b**), and a greater decrease in salivary cortisol levels throughout the day (**c**) when they reported having had time to themselves. Light gray lines represent random slopes, thick black lines represent the fixed slopes. Estimates are based on n = 2299 observations of N = 318 individuals for **a** + **b** and on n = 788 observations of N = 255 individuals for **c**.

personality did not significantly moderate associations between personal time and diurnal cortisol slopes, likely due to reduced power to detect multiple simultaneous cross-level interactions in the smaller cortisol sample. However, in a model with neuroticism entered as a moderator, only, individuals high in neuroticism showed a stronger association between personal time and salivary cortisol (β = −0.08, SE = 0.04, 95% CI = [−0.15, −0.01], p = 0.022; see Supplementary Table 2). Specifically, individuals high in neuroticism experienced a decline in cortisol on days when they reported having had time to themselves (β = −0.14, SE = 0.04, 95% CI = [−0.22, −0.07], p < 0.001), whereas for individuals low in

neuroticism personal time was not significantly associated with diurnal cortisol slope (β = −0.01, SE = 0.04, 95% CI = [−0.09, 0.07], p = 0.811).

**Sensitivity and Follow-Up Analyses**. The results remained consistent after controlling for age, gender, household income, the number of underage children in the household, and the age of the youngest child (for both outcomes), as well as self-rated health and medication use (for cortisol slope as outcome). Therefore, more parsimonious models without covariates are presented. The robustness of findings was also

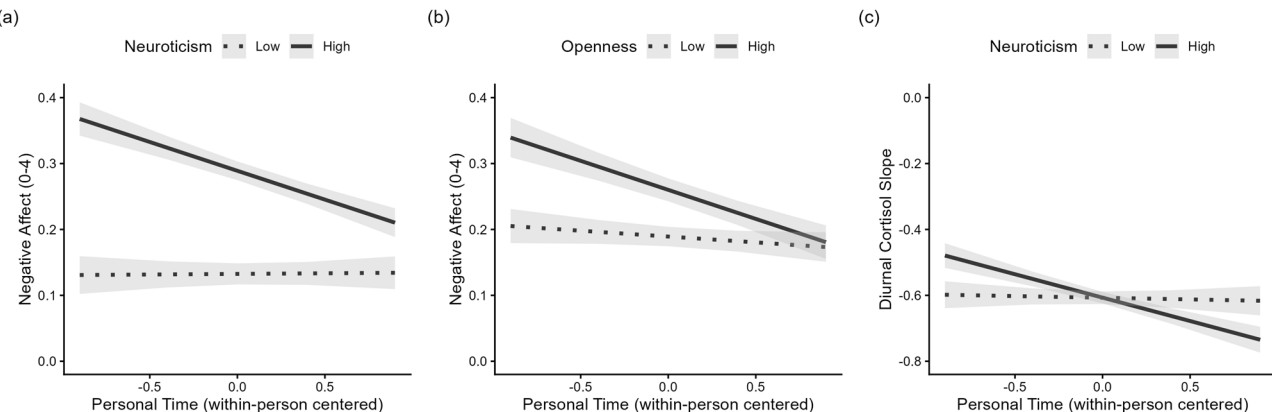

**Fig. 2 | Within-person relationships between personal time, negative affect, and salivary cortisol are moderated by personality.** Individuals high in neuroticism (**a**) and openness (**b**) experienced decreased negative affect on days they had time to themselves, while this association was not significant for individuals low in neuroticism or openness. Additionally, those high in neuroticism showed steeper diurnal cortisol slopes on days with personal time, whereas no significant association was found for individuals low in neuroticism (**c**). The shaded area represents the uncertainty around each fitted line, calculated as fit ± standard error. Estimates are based on $n = 2299$ observations of $N = 318$ individuals for **a** + **b** and on $n = 788$ observations of $N = 255$ individuals for **c**.

tested by excluding participants with low adherence to cortisol sampling ($N = 185$ with ≥3 days of diurnal cortisol slope data; $N = 230$ with ≥2 days). The main effect of personal time on diurnal cortisol slope remained significant across the subsamples, but the interaction with neuroticism reached significance only in the full sample, likely reflecting reduced power.

Participants also reported the number of hours spent on leisure activities each day. Days with opportunity for personal time were associated with more leisure, with participants spending an average of 3.28 h in leisure on personal time days versus 2.29 h on days without personal time. Using a binary indicator, days with personal time co-occurred with days with at least some leisure on 73.8% of measured occasions. When leisure time was used in place of personal time as the predictor, the same pattern of results was observed (see Supplementary Table 3): On days when participants spent more time on leisure activities than usual, they reported higher positive affect ($\beta = 0.05$, SE = 0.01, 95% CI = [0.03, 0.08], $p < 0.001$) and lower negative affect ($\beta = -0.04$, SE = 0.02, 95% CI = [−0.07, −0.00], $p = 0.031$), holding stress exposure constant. Leisure time was also significantly associated with cortisol slope—days with more time spent on leisure were linked to a steeper cortisol decline ($\beta = -0.08$, SE = 0.04, 95% CI = [−0.15, −0.01], $p = 0.034$).

Associations of personal time with affect or cortisol did not significantly differ by gender, age, the age of youngest child, or number of underage children in household (see Supplementary Tables 4–7 for parameters). However, there was a significant three-way interaction between gender, neuroticism, and personal time on negative affect ($\beta = -0.04$, SE = 0.02, 95% CI = [−0.07, −0.00], $p = 0.046$; Supplementary Table 8). Specifically, the reduction in negative affect on days they had time to themselves was even greater for men than for women high in neuroticism, compared to those low in neuroticism (see Supplementary Fig. 1).

## Discussion

As parents, the time cost to time to oneself is estimated to be about 4 h per day for each younger child ≤3 years and 1.5 h per day for each child >3 years[32]. Personal time (time spent free from external demands and available for self-directed activities) can serve a restorative function, helping individuals recover and better prepare for future social and caregiving demands[13]. Thus, it could play an important part for the daily health and well-being of midlife parents who face substantial and ongoing demands related to work, household responsibilities, and children's schedules[29]. Indeed, it was found that parents reported higher levels of positive affect, lower levels of negative affect, and displayed steeper cortisol slopes (indicating better physiological stress recovery) on days on which they had the opportunity to take time for themselves, as compared to days without personal time. These associations were moderated by parents' personality traits, with individuals high in neuroticism and openness experiencing stronger positive correlates of personal time.

## Benefits of personal time for parenting

Participants reported that they had the opportunity to take time for themselves on 4 out of 5 days, on average. Parents might have learned over time to prioritize personal time, recognizing its importance for well-being, or have developed strategies to integrate restorative activities into their daily routines since having their first child. In this study, parents reported higher levels of positive emotions—such as happy, calm, cheerful, and satisfied—and lower levels of negative emotions—such as anger, fear, frustration, and sadness—on days when they had some time for themselves. This is in line with other daily diary research linking personal time to higher vitality, increased positive affect, and decreased negative affect[8,64].

Among the various restorative functions of personal time, previous literature has pointed out its role in emotion regulation[22]. Time to oneself allows space to process difficult problems or feelings in private[65] and has been shown to help down-regulate emotional arousal whether positive or negative[22]. This calming or "deactivation" effect may be especially important for parents of young children, who face a continuous stream of demands, from household responsibilities to emotional and physical caregiving[35]. Thus, it would be interesting for future research to investigate how personal time facilitates emotional down-regulation in different affective contexts, and whether its restorative value differs across life stages (e.g., early parenthood vs. caregiving for aging parents). Related research with middle-aged and older informal caregivers suggests that time for oneself can foster self-connection and enhance well-being—particularly when approached with a mindset of self-kindness[66]. Another important question concerns the dose–response relationship of personal time and well-being—that is, how much is needed, and at what times, to be beneficial? Bradshaw et al.[67] found that even brief periods of time to oneself—as short as 5 min—led to mood improvements and reduced feelings of depletion. Future research should examine whether personal time earlier in the day influences subsequent stress reactivity and emotional recovery. It will also be important to investigate how patterns of personal time across multiple days accumulate and contribute to longer-term health and well-being.

Days of opportunity for personal time were also associated with steeper cortisol slopes, indicating a healthier diurnal cortisol rhythm. In contrast, a flattened cortisol slope (marked by less decline of cortisol levels throughout the day and increased evening levels) is often seen as a physiological indicator of chronic stress, burnout, or poor stress recovery[43,68], suggesting that the body remains in a heightened state of arousal. Dovetailing with this notion, parents report lower levels of perceived stress on days when they

have more time for themselves[64]. In the current study, personal time often coincided with leisure activities, with participants engaging in roughly an hour more leisure on days when they had the opportunity for personal time. For parents, using moments of personal time for self-care—such as napping, listening to soothing music, or engaging in calming and restorative activities like painting or physical exercise—may offer important opportunities for psychological disengagement and emotional recovery from daily pressures[35]. Chronic parental stress has been linked to greater emotional exhaustion, increased risk for internalizing problems such as depression and anxiety, and reduced life satisfaction[30]. These stress-related effects can ripple outward, affecting relationship quality between partners[69] and parenting, ultimately effecting the overall well-being of children[70]. For instance, studies with college students have found that insufficient time alone was associated with elevated levels of trait aggression, anger, and violent tendencies, as well as more aggressive behavior in experimental settings—such as inserting more pins into a voodoo doll[71].

It is also important to recognize that while some personal time may happen in solitude, many parents experience time spent with their children as deeply fulfilling and restorative. In young mothers, time for oneself does not necessarily involve physical separation from the baby; instead, it can coincide with moments of rest and relaxation[35]. In a study of 186 parents of young children, participants tracked their daily activities and rated how much they enjoyed each one[72]. Interestingly, time spent on childcare was linked to greater positive affect than any other activity, including watching TV or cooking.

### Personality differences in benefits of personal time

Associations of personal time with daily negative affect and cortisol slopes were moderated by neuroticism and openness. Specifically, only individuals high (but not those low) in neuroticism showed decreased negative affect and steeper cortisol slopes on days when they had the opportunity to take time for themselves. Neuroticism was also associated with lower overall levels of positive affect and higher overall levels of negative affect throughout the study period. This is in line with previous research demonstrating that individuals high in neuroticism tend to be more irritable, anxious, impulsive, worrisome, tense, fearful, and high-strung—that is, they exhibit lower emotional stability[50].

In the context of parenting, these emotional tendencies may amplify the stress associated with daily caregiving demands[47]. As such, parents high in neuroticism may benefit more from personal time to regulate affect, recover from emotional strain, and prevent overwhelm. Indeed, Ren et al.[52] reported that individuals high in neuroticism perceived the emotion regulation function of time to themselves to be more important, as compared with individuals low in neuroticism. The decrease in negative affect on days with personal time was greater for men than for women high in neuroticism. This aligns with prior research indicating that the link between feelings of time strains for oneself and life satisfaction was stronger for fathers than for mothers[5]. Women's leisure time might be of lower quality than men's because it tends to be more limited, frequently interrupted by caregiving demands, and squeezed into busy schedules rather than planned which could reduce its restorative value[73]. Future research should examine how the context and quality of personal time, not only its quantity, interact with personality to predict daily emotional recovery across genders.

Neuroticism has also been linked to heightened physiological stress reactivity and a greater reliance on passive and less effective coping strategies[74]. While findings on the relationship between neuroticism and HPA axis reactivity remain mixed and somewhat unclear, some evidence suggests that neuroticism is associated with overall higher cortisol output[75]. Personal time may therefore help parents high in neuroticism in down-regulating physiological arousal, potentially leading to lower evening cortisol levels and a steeper diurnal decline on stressful days. Because this study did not include direct measures of momentary physiological arousal or coping responses, it remains unclear whether the benefits of personal time for parents high in neuroticism stem from reduced stress reactivity, cognitive distance, emotional recovery, or simply relief from external demands[45,74]. Future work could strengthen this understanding by incorporating more detailed daily assessments of coping processes, stress reactivity, and additional indices of cortisol regulation (e.g., total cortisol output via area-under-the-curve) to better capture how personal time supports those high in neuroticism.

Openness also moderated the relationship between personal time and negative affect, such that individuals with higher levels of openness experienced a greater reduction in negative affect on days when they had time for themselves. This aligns with existing research linking openness to curiosity, imagination, arts, and intrinsic engagement in self-directed novel activities[50]. People high in openness have been found to value personal time as a means for creativity and self-exploration[52]. These findings suggest that personal time may allow highly open parents to engage in personally meaningful or imaginative activities such as journaling, creative writing, painting, listening to music, or reading fiction[76]. To better understand why personal time is especially beneficial for parents high in openness, future studies could capture concepts such as creativity, absorption and flow experiences[77].

The association between opportunity for personal time and parents' daily well-being was not significantly moderated by conscientiousness, agreeableness, or introversion. This is somewhat unexpected, given that introverts are often characterized as individuals who recharge through solitude and derive energy from spending time alone[49]. Previous research on introversion within the Five Factor Model (as used in this study) has produced mixed results. Some studies have found that introverts show a stronger preference for solitude, engage more frequently in solitary activities, and derive greater enjoyment from them, while others have found no significant differences in the enjoyment of alone time between introverts and extraverts[49,77,78]. It may be important to differentiate between distinct facets of introversion when examining their associations with the benefits of personal time[49]. For example, social introversion—defined as a preference for low levels of social interaction and measured as one dimension of the STAR Introversion Scale[79]—has been found to predict a higher frequency of voluntary solitude episodes[48]. This contrasts with other facets of introversion as measured by the STAR Scale, such as thinking introversion (a tendency toward introspection, imagination, and deep thinking) and anxious introversion (characterized by discomfort or self-consciousness in social settings[79]. Whether time to oneself is actively chosen for its positive qualities or pursued as a way to avoid social interaction has been shown to significantly shape how solitude is experienced[80]; see Rodriguez et al.[81] for a positive reappraisal intervention of solitude. Therefore, distinguishing between anxious introversion and social introversion may be particularly important for future research on the psychological correlates of personal time.

### Strengths, limitations, and future directions

This study offers several key strengths that contribute meaningfully to the literature on personal time and daily well-being. First, the sample was relatively large and demographically diverse, comprising 318 parents recruited through random-digit dialing. The sample included nearly equal numbers of mothers and fathers of underage children, with 19% identifying as non-white, thereby enhancing the generalizability of findings across gender and racial groups. Second, the use of an experience sampling design that incorporated both subjective reports and a physiological marker of stress—salivary cortisol—provides a richer, ecologically valid understanding of parents' daily experiences. Saliva samples were collected in participants' natural environments, capturing diurnal cortisol patterns alongside self-reported experiences of personal time and affect. Finally, the study contributes to a lifespan perspective on time to oneself. While most previous research has focused on time to oneself in childhood, adolescence, or late adulthood, relatively little is known about its role in midlife[82]. By examining personal time in a midlife parenting context, this study helps fill an important gap in understanding how time to oneself functions during a life stage marked by high social and caregiving demands.

Despite these contributions, several limitations should be acknowledged. One key limitation concerns the study's cross-sectional and observational design, which precludes causal conclusions. Although the findings suggest that personal time is associated with better daily well-being, experimental studies are needed to establish whether personal time actively improves stress and affect. Future research could test this by instructing parents to engage in brief periods of personal time (e.g., 15–30 min of intentional time to themselves each day) and assessing changes in subjective well-being and physiological stress markers over time. MIDUS employed a relatively minimal cortisol sampling protocol (four samples per day across 4 days). Future research should include more days of sampling to improve the reliability of diurnal cortisol estimates.

There are also limitations related to sampling. The study was conducted in the United States, and it remains unclear how the findings generalize to parents in other cultural contexts. Cultural norms surrounding personal time likely differ depending on dominant self-construals: in independent cultures, time to oneself may be more valued and accessible, yet social networks may be weaker, potentially increasing vulnerability to loneliness[83]. Conversely, collectivistic contexts may offer stronger social support and time to oneself might be viewed as meaningful opportunity for introspection, reflection, and self-cultivation[84]. Jiang et al.[85] found that exposure to Chinese culture, whether through ethnicity or residence, was associated with better affective experiences during solitude. Supporting this, Japanese participants reported more favorable beliefs about being alone than their American counterparts, and these beliefs were associated with lower levels of loneliness in both cultural groups[84]. Future work should investigate how cultural values, including interdependent/dependent self-construals, shape the meanings and outcomes of personal time in daily life.

Measurement limitations should also be noted. Personal time was assessed using a single binary item asking whether participants had the opportunity to take time for themselves on a given day (yes/no), which may oversimplify the complexity of experiences of personal time. Evidence from prior experience sampling studies suggests that personal time and solitude fluctuate substantially from day to day in response to situational demands while also exhibiting modest trait-like stability across individuals[8,86]. Future research should employ more nuanced measures that capture the different types and qualities of personal time, as well as its interplay with related concepts. While similar patterns were observed when using hours spent on leisure activities as a predictor, the daily interplay between personal time and related constructs such as solitude, privacy, and leisure remains to be explored.

This study did not assess what specific activities parents engaged in during their time to themselves, making it difficult to determine whether benefits are driven by personal time per se or by the activities undertaken during that time. For instance, whether personal time involves technology use, physical separation from others, or internal focus may significantly influence whether personal time is experienced as restorative[9]. Moreover, personal time used for passive entertainment like watching Netflix or social media, recreational activities like going for a run, contemplative activities like planning or reflecting, or not doing any activity at all (activity-less personal time) may have differing effects on daily well-being[49].

Another important yet overlooked factor is control over personal time. For many parents—especially those caring for young children—time for oneself is not only limited but unpredictable[35]. For instance, mothers may only find time for themselves during naps, the length of which is difficult to anticipate. This lack of control can lead to frustration when personal time is interrupted or insufficient for meaningful rest or engagement[35]. Finally, future research should also include other, related personality constructs such as dispositional autonomy (acting in accordance with one's authentic interests and values) and sensory processing sensitivity (a heightened responsiveness to environmental and emotional stimuli[48,51].

## Conclusion

This study leveraged an experience sampling design—the gold standard for capturing in vivo experiences—and drew on a subsample of the MIDUS (Midlife in the United States) study, a nationally representative longitudinal panel focused on health, well-being, and aging in midlife and beyond. Findings underscore the potential psychological and physiological benefits of personal time: on days when midlife parents had the opportunity to take time for themselves, they reported reduced negative affect and showed steeper cortisol slopes, indicating better stress recovery. Notably, these associations were moderated by personality, with individuals high in neuroticism and openness experiencing the strongest benefits. For parents, periods of personal time may constitute their primary—if not only—opportunity to engage in leisure pursuits, such as reading for pleasure, exercising, engaging in creative hobbies, or simply resting without caregiving demands. Given the significant time demands of parenting, personal time may serve as a valuable and often overlooked resource for supporting emotional and physiological well-being in daily life.

## Data availability

The data utilized in this study (MIDUS Refresher 1 Daily Diary) are publicly available through the Inter-university Consortium for Political and Social Research (ICPSR) at https://www.icpsr.umich.edu/web/ICPSR/studies/37083.

## Code availability

Analytic code is available on the OSF.

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

## Acknowledgements

Theresa Pauly gratefully acknowledges support from the Canada Research Chairs Program. Publicly available data was used for this research, provided by the longitudinal study titled "Midlife in the United States", (MIDUS) managed by the Institute on Aging, University of Wisconsin. Since 1995, the MIDUS study has been funded by the following: John D. and Catherine T. MacArthur Foundation Research Network; National Institute on Aging (P01-AG020166); National Institute on Aging (U19-AG051426). Biomarker data collection was further supported by the NIH National Center for Advancing Translational Sciences (NCATS) Clinical and Translational Science Award (CTSA) program as follows: UL1TR001409 (Georgetown); UL1TR001881 (UCLA); 1UL1RR025011 (UW). The author received no specific funding for this work. Theresa Pauly gratefully acknowledges support from the Canada Research Chairs Program. The funders had no role in study design, data collection and analysis, decision to publish or preparation of the manuscript.

## Competing interests

The author declares no competing interests.
