## [Transparent Peer Review file · Communications Psychology]

Personality Moderates Associations Between Personal Time and Parental Well-being

Corresponding Author: Dr Theresa Pauly

Version 0:

Decision Letter:

Dear Dr Pauly,

Thank you for your patience during the peer-review process. Your manuscript titled "'No Time for Myself': Personality Moderates Associations Between Positive Solitude and Parental Well-being" has now been seen by 3 reviewers, and I include their comments at the end of this message. They find your work of interest but raised some important points. We are interested in the possibility of publishing your study in Communications Psychology, but would like to consider your responses to these concerns and assess a revised manuscript before we make a final decision on publication.

We therefore invite you to revise and resubmit your manuscript, along with a point-by-point response to the reviewers. Please highlight all changes in the manuscript text file.

Editorially, we consider it important that the revised manuscript fully address the conceptual and methodological concerns raised by the reviewers. In particular, the conceptualization and operationalization of the study variables, especially positive solitude, should be revised with better integration into existing literature.

Please ensure you follow our statistical guidelines when reporting statistics (<https://www.nature.com/commpsychol/submit/submission-guidelines#statistical-guidelines>). Please note in particular our requirements for the reporting and interpretation of null-results. Non-significant findings derived from null-hypotheses significance tests should be reported in full, but may not be interpreted. Where you interpret null results, this interpretation must be based on Bayes Factors or equivalence tests.

I am attaching an Editorial Requests Table that details critical reporting requirements for the revised manuscript. Please attend to each item and ensure your manuscript is fully compliant. If your revised manuscript is not aligned with these requests on major issues, such as those concerning statistics, it may be returned to you for further revisions without re-review.

Please submit the following items:

- Revised manuscript
- Point-by-point response to the referees' comments
- Cover letter (as a separate document)
- <https://www.nature.com/documents/nr-reporting-summary.pdf>>Nature Research Reporting Summary
- Completed Editorial Request Table (attached).

via this link: Link Redacted .

Additional guidance is available in our style and formatting guide Communications Psychology formatting guide.

Best regards,

Jennifer Bellingtier

Jennifer Bellingtier, PhD
Senior Editor
Communications Psychology

REVIEWER EXPERTISE:

Reviewer #1 well-being, daily life, stress

Reviewer #2 solitude, well-being

Reviewer #3 family relationships, stress

REVIEWER REPORTS:

Reviewer #1 (Remarks to the Author):

The current paper, "No time for myself": Personality moderates associations between positive solitude and parental well-being" was an interesting examination of daily positive solitude and the association with negative affect and cortisol in parents of underage children. Given the nuance of positive solitude, this manuscript can provide an innovation and extension to the current literature. Strengths include the use of a large national dataset with eight days of diary data, 4 assessments per day for cortisol, and thoughtful consideration of the population. Conclusions drawn from the results are tapered to limitations and are not overinterpreted. Although strengths, a handful of limitations decrease this reviewer's enthusiasm for the manuscript. Below are both major and minor considerations.

Major

1. The abstract notes "wake-evening slopes" as the main outcome for cortisol; however, this is not a term that the current reviewer is familiar with. More information may be needed to better articulate what this indicator is and how it differs from commonly utilized terms: cortisol awakening responses, diurnal cortisol slopes, dynamic range, area under the curve. This can be brought through in the section on cortisol on page 7. Notably, the author mentions flatter diurnal cortisol slopes in previous research but does not specify whether this is the phenomena examined in the current study until the hypothesis. It may be pertinent to include rationale as to why the authors are only interested in the DCS, rather than CAR, AUC, or dynamic range, which, with the four data points could be viable measures to operationalize in MIDUS.
2. The current reviewer learned a lot about positive solitude in the introduction; however, one question that remained was how to measure positive solitude. In several places (e.g., page 4 discussing positive experiences and positive solitude), there was a different general definition of the concept than the first operationalization (i.e., time to oneself). As such, it's unclear if positive solitude refers to spending time alone, positive activities that one engages in when alone, or activities done alone to promote positive emotions. This may be addressed easily with some editing of the text to make it clear that positive solitude is associated with activities and emotions, but that the measurement is focused on whether time was spent alone that day.
3. The author makes a strong case for gender differences in parental time; it stands to reason, especially with the near 50-50 split in gender for the sample, that examining gender as a moderator is a pertinent avenue to examine exploratorily. Why did the author not explore this more fully?
4. Did the author consider examining positive affect in conjunction with negative affect? Especially given the restorative

properties mentioned in the introduction, it would be interesting to see if negative affect decreased and low arousal positive affect increased on days when solitude occurred. This may also add to the evidence that the solitude participants were experiencing was positive.

5. The current reviewer is concerned about the cortisol sampling; although the author has no ability to reconcile this (as they are utilizing secondary data), Segerstrom and colleagues (2014) note that 3 days is the minimum number of days that should be used to determine cortisol slopes between persons and 3-4 days is needed to calculate DCS reliably. Given this, anything less than 3 days per person would not provide a reliable estimate of the cortisol reported. How many of the sample mentioned (N=255) had at least 3 days of valid data? Moreover, it may be useful to run sensitivity analyses with the subsample with 3+ days to determine if the current results are robust to the loss of -3 days of data.

6. The author discusses the results pertaining to cortisol but does not discuss why models with all personality traits vs. just neuroticism resulted in a lack of association. Why might one see differential associations depending on personality inclusion; further, did the author examine other personality traits on their own, or just neuroticism?

Minor

1. Based on the abstract definition of positive solitude, it is unclear how positive solitude differs from negative solitude - which is noted as a potential (e.g., enriching and challenging; wide range of emotional states) valence for solitude.

2. The author makes a good transition / case to argue for the importance of positive solitude for people parenting especially in the first year after childbirth; however, the MIDUS sample is on average 7-8 years old. Can a similar case be made for parenting across childhood broadly? Notably, given the introduction, the current reviewer was anticipating early childhood (i.e., 0-6).

3. It is unclear how the current research addresses the long-term ramifications of stress on physical health (page 7). Notably, the cortisol patterning is not reported on long-term and has large variability across 4 days of data; if associations were over 10 years in MIDUS, it could be argued, but this is not the case.

4. The author may mean random digit dialing rather than random dial digits in the procedures and participants on page 9.

5. What is meant by the following sentence, "One person was missing information on personality, resulting in a sample size of n = 2,299 surveys from N = 318 participants."? Do the authors mean that there were 2,299 daily diary assessments available from the 318 participants?

6. How many completed days (7.23 out of 8) were days with cortisol sampling?

7. I appreciate the inclusion of the statistical power for the main analyses, but given the cortisol analyses are with 255 participants, did the author run the same power analysis with the slightly smaller sample size?

8. I am surprised that the participants reported positive solitude so often - could this be because they, on average, have older children (and have had 7 or so years to adjust to daily schedules)?

9. In the results section, it is unclear why the author is reporting on daily stressor exposure. Although interesting, it is not a hypothesized pathway; justification in the introduction should be added, or this should be dropped.

10. Out of curiosity, is it possible to determine how often solitude and leisure time co-occurred? In the discussion the author mentioned solitude and self-care activities that were similar to leisure activities, so it would be curious to know.

Reviewer #2 (Remarks to the Author):

Thank you for the opportunity to review this paper. Overall, I think this is an interesting study, and I want to commend the researcher for sharing their code and for explaining the analyses and results very clearly. The results are also presented in a way that is easy to follow.

That said, my main concern is with the framing of the research.

1. I am not convinced the study is really about solitude. The measure used—"Since this time yesterday, did you have the opportunity to take time for yourself?"—does not necessarily capture solitude. Prior work, including Nguyen et al. (2025)'s work on mothers, shows that "personal time" can include self-care or enjoyable activities, sometimes even bonding with one's baby. In other words, "time for yourself" does not always mean being alone. This is why I think the paper should be framed around "time for yourself" (or "me-time") rather than solitude. Solitude may be part of it, but the construct as measured here is broader.

Also, here is the latest reference! Nguyen, T. T., Konu, D., Tetteh, D., Tshimbalanga, P., Weissová, J., & Xiong, M. (2025). "I got all sorts of solitude, but that solitude wasn't mine": A Mixed-Methods Approach to Understanding Aloneness during Becoming a Mother. *Journal of British Psychology*. Doi: <https://doi.org/10.1111/bjop.70019>

2. I suggest restructuring the Introduction. It would work better if it started with the literature on parents and the complexity of their solitude, then discussed research on solitude while still emphasized that, for parents, it is about "me-time", and finally personality differences. As pointed out in #1, it is important that you distinguish between solitude and "me-time". This distinction should be made clear in the Introduction. Otherwise, there's a risk of muddling the solitude literature.

3. Similarly, I don't think the item fully captures positive solitude (which is solely about solitude) as it has been conceptualized in works like Weinstein et al. or Ost-Mor et al. "Me-time" overlaps with positive solitude but for parents, can capture bonding time with baby or social time when someone else take over childcare to leave parents some space. See Nguyen et al. (2025)'s Discussion section.

4. The Discussion should also be revised to reflect this reframing. Right now, equating "time for yourself" with positive solitude is problematic, especially without confirmation that participants who said yes were in fact alone and not interacting with others.

5. In addition, I would like to request some clarification about the sample and descriptive data. Specifically, could the authors provide more information on how healthy or stressed this sample of parents was? This would help clarify, especially because I wonder if the sample may have more support, given that 79% of days were coded as "yes" for having time to

themselves. Since there are relatively fewer data points for “no” time to themselves, how does this imbalance affect the generalizability of the results? Overall, I would like to know a bit more about this group of parents and how representative they might be.

I think this paper addresses an important and timely issue. The strengths are the open sharing of code, the clear explanation of analyses, and the accessible presentation of results. With reframing and some additional detail on the sample, this paper could make a strong contribution.

Reviewer #3 (Remarks to the Author):

This study examined how parent solitude is associated with parent negative affect and cortisol (an indicator of stress). Overall the study is well-written, and the topic is interesting and well-developed. More information is needed throughout the differentiate terms, and important methodological information and additional analyses should be added. Suggestions are also given regarding streamlining the introduction to focus more specifically on study findings and not extend beyond the scope of this paper.

Abstract:

Add child ages to the abstract.

Introduction

It would strengthen the paper to have clearer definitions throughout, as several of the studies reviewed looked at solitude but some at related concepts (such as time alone). For example, the introduction states that solitude is different than being alone, yet throughout the paper discussion of solitude refers to the benefits of time alone. More is also needed to understand how solitude relates to privacy (and how privacy is defined).

How much does solitude matter vs what you do during solitude? It seems important to make this distinction throughout the introduction, as studies reviewed look at multiple related concepts, and clearly distinguishing would enhance the paper. It would also enhance the paper to discuss the implications of these distinctions for understanding the strengths/gaps of the literature as a whole.

Are there other ESM studies that have examined solitude? What do we know from prior work about how much it varies from day to day? And if is a trait?

Dysregulated cortisol can also lead to cortisol hyperarousal, and this should be discussed as well in the introduction.

It would be helpful to add more about burnout when you are discussing the experiences of parents and stress, especially as you cite burnout literature when discussing personality. These are distinct concepts and the nuances of their differences should be made clear when reviewing the literature.

More explicit hypotheses on specific personality traits would enhance the paper.

Check reference formatting (minor)

Methods

Much more information is needed in the methods section. How often and when were individuals in the refresher sample surveyed? Does the refresher sample differ from the full sample? How was the daily diary data collected? What time of day was it collected?

The measure of positive solitude should be described more in relation to the literature. From the question asked in the survey, this seems to get more about being alone than necessarily solitude. And there is nothing in the question as I read it, that gets at whether or not that solitude is positive or negative. It seems to get at time alone only.

More information about cortisol measures and analysis should be added here. Specifically, more information is needed on collection and processing should be added here (rather than referring to another paper only). Was the data log transformed- and if so how? How many people provided full cortisol samples and both samples needed to calculate the score for this paper? Did people dropped for noncompliance differ from those that were compliant? Why did you divide your score of the slope by the time elapsed between the samples?

It would strengthen the paper to have a data analysis plan. I had many questions as I was reading the methods section.

Some of them were answered later (and some not). It would enhance the paper to discuss your steps in detail earlier in the paper. For example, more information is needed earlier about the other control variables that you also tested, but then later dropped (age, sex, etc.). If you planned to test the moderators separately or together, etc.

Did you control for medications known specifically to interfere with cortisol?

How many individuals were dropped due to having too much missing data for the personality measures? Were the personality measures added as level two predictors only?

What was the racial makeup of your sample (beyond what % was white)?

Results

Table 2 would be strengthened by labeling the within and between person effects more clearly

It is unclear if the moderators were tested individually? And while controlling for the other personality traits? Or if all the moderators were tested in the same model? And did the authors also test moderation for the between-person effects?

The sensitivity and follow up analyses are interesting, and some of it would enhance the paper and I recommend moving some of it and expanding it throughout. For example, given that solitude's effects may be solitude and what you do with that time, adding leisure time would enhance the paper. Adding the lagged models to the main body of the paper would also strengthen it. The question on daily stressors as a potential moderator seems less relevant to the overall question of this paper, and I recommend dropping it.

Discussion

The discussion would benefit from being more streamlined around the main questions of this paper.

The discussion goes beyond the main analysis of this paper (in that it discusses what individuals do with solitude, which was not assessed here). However, adding the leisure findings more thoroughly would enhance this, and expand the ability of authors to discuss this in the discussion. Also, the authors discuss potential downsides of solitude, but findings do not support this- so be sure to discuss those in context specifically of what you find. And how your findings support other work or inspire future research questions based on gaps in formation that you had or contradictions with other studies.

Similarly it would strengthen the paper to streamline the discussion of personality differences, and focus it more on the questions asked here and how these are/are not supportive of other literature and possible explanations and things you could not answer but would be next steps.

The limitations section was very thoughtful and thorough.

If you experience problems in linking your ORCID, please contact the Platform Support Helpdesk.

Version 1:

Decision Letter:

Dear Dr Pauly,

Your manuscript titled "'No Time for Myself': Personality Moderates Associations Between Positive Solitude and Parental Well-being" has now been seen by our reviewers, whose comments appear below. In light of their advice I am delighted to say that we are happy, in principle, to publish a suitably revised version in Communications Psychology.

We therefore invite you to revise your paper one last time to address the remaining concerns of our reviewers and a list of editorial requests. At the same time we ask that you edit your manuscript to comply with our format requirements and to maximise the accessibility and therefore the impact of your work.

EDITORIAL REQUESTS:

SUBMISSION INFORMATION:

OPEN ACCESS:

Communications Psychology is a fully open access journal. Articles are made freely accessible on publication. For further information about article processing charges, open access funding, and advice and support from Nature Research, please visit <https://www.nature.com/commpsychol/open-access>

* DATA AVAILABILITY:

All Communications Psychology manuscripts must include a section titled "Data Availability" at the end of the Methods section. More information on this policy, is available in the Editorial Requests Table and at <http://www.nature.com/authors/policies/data/data-availability-statements-data-citations.pdf>

Link Redacted

Best regards,

Jennifer Bellingtier

Jennifer Bellingtier, PhD
Senior Editor
Communications Psychology

REVIEWER EXPERTISE:

Reviewer #1 well-being, daily life, stress

Reviewer #2 solitude, well-being

Reviewer #3 family relationships, stress

REVIEWERS' COMMENTS:

Reviewer #1 (Remarks to the Author):

I commend the author for the massive and thorough rebuttal of all reviewer comments. For only one author, this can be quite an undertaking. I have a handful of points based on the revised manuscript; however, all are minor.

1. I appreciated the reminder to reviewer 2's point that 21% of days with no personal time still allows for an examination of the questions at hand. I wonder if adding two things would strengthen this point. Notably, with an eight-day diary study this would mean that ~3 days were no personal time (on average) and ~5 were with personal time. Although unbalanced, that still provides plenty of days with and without personal time. The other piece could be to clarify (perhaps for early career readers without the statistical understanding) that you ran unconditional/empty models to identify the between- and within-persons variance is across the sample for your variables (i.e., page 16) which indicated that 75% of the variance of personal time, for example, is at the day-to-day level (or within-persons).

2. Thank you for the additional analyses throughout to answer our questions/comments. I am unsurprised that leisure time and time alone are associated with one another - and I agree that especially for parents, alone time may be the only time that

they can engage in leisure activities.

3. Could you clarify how the Refresher daily diary project had an N of 782 but the final analytic sample was N = 318? Based on what's written, it seems like only 318 people had an underage child living in the household, but it wasn't entirely clear.

Reviewer #2 (Remarks to the Author):

Thank you for your thorough revision. It is great that you have reframed the work so it more aligns with the construct "time for oneself" that you measured. I also thought it was nice to see more statistics around what those "yes" response to having time for oneself means. Your revision has satisfied my concerns.

Reviewer #3 (Remarks to the Author):

The authors have addressed most of my concerns. I appreciate the improved conceptual clarity and thoroughness of this paper and expect it will make an important contribution to the field. I have a few minor suggestions for further revision.

I appreciate the focus on personal time and conceptual clarity. Given your measure captures opportunities for personal time (as opposed to whether they actually had personal time)- I would recommend capturing this nuance throughout. For example by specifying opportunities for personal time in the abstract, results, discussion, limitation.

I appreciate your reasons not to have hypotheses about personality- it may be helpful to briefly discuss these reasons in the paper.

It would be helpful to specify the underlying methods used to back up statements of sample differences.. For example "this sample of parents was significantly younger, had higher average household income, better self-rated health, less time to themselves, lower average negative affect, and steeper cortisol slopes than the remainder of the Refresher Daily Diary sample—give some more information to back this up" and "Those excluded due to non-compliance did not differ from included participants in terms of sex, ethnicity, age, subjective health, average personal time, or positive or negative affect, but they did report significantly lower income"

Also, it would be helpful to specify your methods for follow-ups to derive estimates for those high/low in neuroticism (I'm assuming simple slopes were tested, specify the process). For example, here Figure 2, individuals high in neuroticism ($\beta = -0.09$, $SE = 0.02$, $p < .001$) and in openness ($\beta = -0.09$, $SE = 0.02$, $p < .001$) experienced a significant decrease in negative affect on days when they reported having had time to themselves. This was not true for individuals low in neuroticism ($\beta = 0.00$, $SE = 0.02$, $p = .937$) or openness ($\beta = -0.02$, $SE = 0.02$, $p = .408$).

Reviewer 1's comments:

The current paper, “No time for myself: Personality moderates associations between positive solitude and parental well-being” was an interesting examination of daily positive solitude and the association with negative affect and cortisol in parents of underage children. Given the nuance of positive solitude, this manuscript can provide an innovation and extension to the current literature. Strengths include the use of a large national dataset with eight days of diary data, 4 assessments per day for cortisol, and thoughtful consideration of the population. Conclusions drawn from the results are tapered to limitations and are not overinterpreted. Although strengths, a handful of limitations decrease this reviewer’s enthusiasm for the manuscript. Below are both major and minor considerations.

Response: I thank the reviewer for highlighting the strengths of the current paper and address all comments below.

Major comments.

1. The abstract notes “wake-evening slopes” as the main outcome for cortisol; however, this is not a term that the current reviewer is familiar with. More information may be needed to better articulate what this indicator is and how it differs from commonly utilized terms: cortisol awakening responses, diurnal cortisol slopes, dynamic range, area under the curve. This can be brought through in the section on cortisol on page 7. Notably, the author mentions flatter diurnal cortisol slopes in previous research but does not specify whether this is the phenomena examined in the current study until the hypothesis. It may be pertinent to include rationale as to why the authors are only interested in the DCS, rather than CAR, AUC, or dynamic range, which, with the four data points could be viable measures to operationalize in MIDUS.

Response: In the cortisol literature, some researchers have computed the diurnal cortisol slope as (1) the decline in cortisol levels from the wake-up sample to the evening sample, (2) the decline from the cortisol awakening response to the evening sample or (3) the estimated linear effect of time of day using all samples collected on a given day. By using the term “wake–evening cortisol slope,” I intended to clarify that I adopted the first approach, which is most commonly used in the field (e.g., 64% of studies in Adam et al.’s 2017 meta-analysis). That said, I agree with the reviewer that “diurnal cortisol slope” is the more widely recognized term, and I have updated the text accordingly.

Many researchers exclude the CAR when calculating the diurnal slope because it is regulated by distinct biological mechanisms (Clow et al., 2010; Adam et al., 2015) and often shows weaker associations with health outcomes compared to the wake–evening slope (Adam et al., 2017). I chose the wake–evening slope rather than regressing cortisol levels on time using all daily samples because the former is less sensitive to the influence of daytime outliers (e.g., unusually high afternoon values). I have now clarified the reasoning behind that choice in the revised manuscript:

(p. 13): “The wake–evening cortisol slope is the most commonly used marker of diurnal cortisol decline and less vulnerable to distortion from daytime outliers, as compared to regression-based estimation methods that use all daily samples³⁸.”

I also outline why I focus on diurnal cortisol slopes, specifically:

(p. 9): “This study focuses on diurnal cortisol slopes as an index of daily stress physiology. Unlike other cortisol measures (e.g., area-under-the-curve, awakening response), a flattened slope specifically reflects reduced recovery, indicating impaired down-regulation of cortisol by evening ⁴⁸.”

2. The current reviewer learned a lot about positive solitude in the introduction; however, one question that remained was how to measure positive solitude. In several places (e.g., page 4 discussing positive experiences and positive solitude), there was a different general definition of the concept than the first operationalization (i.e., time to oneself). As such, it's unclear if positive solitude refers to spending time alone, positive activities that one engages in when alone, or activities done alone to promote positive emotions. This may be addressed easily with some editing of the text to make it clear that positive solitude is associated with activities and emotions, but that the measurement is focused on whether time was spent alone that day.

Response: Reviewer 2 had a similar concern pertaining to conceptual clarity given that the measure used asked “*Since this time yesterday, did you have the opportunity to take time for yourself?*”. They suggested to frame the manuscript around “personal time” and “time for oneself” rather than solitude. While solitude may be one aspect of personal time, the construct in this study encompasses a broader category. I have therefore revised the manuscript (particularly referring to more literature on the well-being implications of personal and leisure time in the introduction) to reflect this change. Examples are:

(p. 3) “Many adults report feeling short on personal or leisure time^{1,2}. A subjective feeling of lack of personal time has been associated with diminished well-being and lower quality of life ^{3,4}. Time strains are particularly pronounced for midlife parents, who are often juggling multiple roles at once—managing family responsibilities, caring for aging parents, and working in high-demand career stages ^{5,6}. This study explores the role of personal time for the daily emotional and physiological well-being of parents of underage children, and examines whether these associations are moderated by personality.”

(p. 3): “Personal time refers to time that an individual spends in self-directed, restorative or voluntary activities, free from obligatory work, caregiving, or household duties ^{7,8}. Personal time can but does not necessarily need to happen when alone, defined as the physical absence of other people, or in solitude, defined as the lack of social interaction ⁹.”

(pp. 3-4): “In the life-balance literature, research indicates that a three-dimensional model—including work time, social time, and personal time—better predicts health outcomes than a two-dimensional model based only on work and social time ¹³. Research has linked engaging in leisure activities or activities that are driven by intrinsic motivation with enhanced recovery, defined as the process to replenish resources like energy, attention, and mood when not exposed to further demands ^{21,22}”

3. The author makes a strong case for gender differences in parental time; it stands to reason, especially with the near 50-50 split in gender for the sample, that examining gender

as a moderator is a pertinent avenue to examine exploratorily. Why did the author not explore this more fully?.

Response: I appreciate the reviewer's comment. I examined whether gender moderated the associations between daily personal time and both negative affect and diurnal cortisol slopes, and found no significant overall gender differences (described on p. 19 in the sensitivity and follow-up analyses). However, I did observe that the reduction in negative affect on days with more personal time was stronger for men than for women high in neuroticism, consistent with prior work showing stronger links between perceived time for oneself and life satisfaction among fathers compared to mothers. I have now incorporated this point into the discussion:

(p. 22-23): "The decrease in negative affect on days with personal time was greater for men than for women high in neuroticism. This aligns with prior research indicating that the link between feelings of time strains for oneself and life satisfaction was stronger for fathers than for mothers⁵. Women's leisure time might be of lower quality than men's because it tends to be more limited, frequently interrupted by caregiving demands, and squeezed into busy schedules rather than planned which could reduce its restorative value⁷³. Future research should examine how the context and quality of personal time, not only its quantity, interact with personality to predict daily emotional recovery across genders."

4. Did the author consider examining positive affect in conjunction with negative affect? Especially given the restorative properties mentioned in the introduction, it would be interesting to see if negative affect decreased and low arousal positive affect increased on days when solitude occurred. This may also add to the evidence that the solitude participants were experiencing was positive.

Response: In response to the reviewer's comment, I have added positive affect as a third outcome to the statistical models. As the reviewer suggested, positive affect increased on days when participants were able to take time for themselves (see revised Table 2, Figure 1, and results on pp. 16-17).

5. The current reviewer is concerned about the cortisol sampling; although the author has no ability to reconcile this (as they are utilizing secondary data), Segerstrom and colleagues (2014) note that 3 days is the minimum number of days that should be used to determine cortisol slopes between persons and 3-4 days is needed to calculate DCS reliably. Given this, anything less than 3 days per person would not provide a reliable estimate of the cortisol reported. How many of the sample mentioned (N=255) had at least 3 days of valid data? Moreover, it may be useful to run sensitivity analyses with the subsample with 3+ days to determine if the current results are robust to the loss of -3 days of data.

Response: I appreciate the reviewer's concern regarding cortisol data reliability. In our sample, 230 of 255 participants provided at least two days of DCS data, and 185 provided at least three days. The main effect of personal time on cortisol slope remained significant and consistent across models using the full sample (N = 255) as well as the subsets (N = 230 and N = 185). However, the interaction between personal time and neuroticism predicting

DCS was only significant in the full sample model. Given that power to detect cross-level interactions in multilevel models is strongly dependent on Level-2 sample size, this likely reflects reduced power rather than a true absence of effect. I have added this point to the sensitivity analysis section and now note in the strengths and limitations that future studies should collect more days of cortisol data.

(p. 18): “The robustness of findings was also tested by excluding participants with low adherence to cortisol sampling (N = 185 with ≥ 3 days of diurnal cortisol slope data; N = 230 with ≥ 2 days). The main effect of personal time on diurnal cortisol slope remained significant across the subsamples, but the interaction with neuroticism reached significance only in the full sample, likely reflecting reduced power.”

(p. 25): “MIDUS employed a relatively minimal cortisol sampling protocol (four samples per day across four days). Future research should include more days of sampling to improve the reliability of diurnal cortisol estimates.”

6. The author discusses the results pertaining to cortisol but does not discuss why models with all personality traits vs. just neuroticism resulted in a lack of association. Why might one see differential associations depending on personality inclusion; further, did the author examine other personality traits on their own, or just neuroticism?

Response: As prompted by your feedback below, I re-examined statistical power for the full sample versus the cortisol sample. Although power in the full affect sample (N = 318) exceeded 90% for both within-person associations and cross-level interactions, power was more limited for models predicting diurnal cortisol slope. With 255 participants providing four days of cortisol data, power was estimated at 71% for detecting within-person effects and 61% for cross-level interactions. Thus, only for the cortisol outcome do I believe that reduced power may explain why the interaction was not significant when all five personality traits were included simultaneously. Supporting this interpretation, the interaction between personal time and neuroticism approached .05 when controlling for the other traits ($p = .0518$), but was statistically significant when entered alone ($p = .0235$). When examined individually, none of the other personality traits significantly moderated associations between personal time and diurnal cortisol slope.

(p. 18): “When all personality traits were entered into the same model, personality did not moderate associations between personal time and diurnal cortisol slopes, likely due to reduced power to detect multiple simultaneous cross-level interactions in the smaller cortisol sample. However, in a model with neuroticism entered as a moderator, only, individuals high in neuroticism showed a stronger association between personal time and salivary cortisol.”

Minor comments

1. Based on the abstract definition of positive solitude, it is unclear how positive solitude differs from negative solitude - which is noted as a potential (e.g., enriching and challenging; wide range of emotional states) valence for solitude.

Response: Thank you for this comment. In line with your feedback and Reviewer 2's suggestion for greater conceptual clarity, I have reframed the manuscript to focus on

“personal time” or “time to oneself,” which represents a positive valenced construct and avoids ambiguity surrounding positive versus negative forms of solitude. I included a definition of personal time in the revised introduction:

(p. 3): “Personal time refers to time that an individual spends in self-directed, restorative or voluntary activities, free from obligatory work, caregiving, or household duties ^{7,8}”

2. The author makes a good transition / case to argue for the importance of positive solitude for people parenting especially in the first year after childbirth; however, the MIDUS sample is on average 7-8 years old. Can a similar case be made for parenting across childhood broadly? Notably, given the introduction, the current reviewer was anticipating early childhood (i.e., 0-6).

Response: Thank you for this helpful clarification. I agree that the introduction placed more emphasis on the early postpartum period. To better align our theoretical framing with the MIDUS sample—who are, on average, parenting school-aged children—I have revised the introduction to emphasize that time strains remain high throughout childhood. Parents continue to juggle substantial caregiving, work, and household responsibilities well beyond the first years of life, and midlife is often a peak period of competing time demands. I have now clarified that personal time may remain scarce and valuable across the broader parenting years, not solely in infancy.

(p. 3): “Time strains are particularly pronounced for midlife parents, who are often juggling multiple roles at once—managing family responsibilities, caring for aging parents, and working in high-demand career stages ^{5,6}.”

(p. 5): “Parenting continues to impose substantial and often intensifying demands throughout childhood and adolescence. Midlife parents frequently manage peak career responsibilities while coordinating children’s school schedules, extracurricular activities, and evolving socioemotional needs, leaving little discretionary time for personal recovery ²⁸.”

(p. 19): “Personal time can serve a restorative function, helping individuals recover and better prepare for future social and caregiving demands ¹². Thus, it could play an important part for the daily health and well-being of midlife parents who face substantial and ongoing demands related to work, household responsibilities, and children’s schedules.”

3. It is unclear how the current research addresses the long-term ramifications of stress on physical health (page 7). Notably, the cortisol patterning is not reported on long-term and has large variability across 4 days of data; if associations were over 10 years in MIDUS, it could be argued, but this is not the case.

Response: In a meta-analysis examining diurnal cortisol slopes (DCS) and health outcomes, Adam et al. (2017) found that collecting more days of cortisol data did not substantially increase the strength of associations with mental and physical health outcomes. This is notable given the considerable state-related variability in DCS; it is

therefore striking that even single-day measures of DCS have been linked to negative health outcomes. The presence of these associations despite high day-to-day variability suggests that the “signal” of dysregulated slopes is significant. I assume that flattened or weakened cortisol slopes that happen in a day-to-day dynamic reflect poorer recovery, which, if persistent over time, could contribute to maladaptive HPA axis regulation. Such dysregulation, in turn, may lead to adverse health outcomes. While the current study does not include a long-term health follow-up, these findings provide a conceptual basis for how daily variations in recovery could accumulate to affect long-term physical health. I have now outlined this rationale more clearly in the introduction:

(p. 7): “Overall, flatter diurnal cortisol slopes have been linked to a wide range of adverse mental and physical health outcomes, including increased risk for depression, impaired immune functioning, obesity, cancer, and cardiovascular disease^{39,40}. Even single- or few-day measures of flattened diurnal cortisol slopes are linked to negative health effects, suggesting that sustained reductions in physiological recovery may gradually lead to maladaptive HPA axis regulation³⁹.”

4. The author may mean random digit dialing rather than random dial digits in the procedures and participants on page 9.

Response: Thanks, I have corrected this sentence (p. 9: “Participants were recruited through random digit dialing.”)

5. What is meant by the following sentence, “One person was missing information on personality, resulting in a sample size of $n = 2,299$ surveys from $N = 318$ participants.”? Do the authors mean that there were 2,299 daily diary assessments available from the 318 participants?

Response: I have now clarified that there were a total of 2299 surveys available across participants:

(p. 10): “One person was missing information on personality, resulting in a sample size of $N = 318$ participants who provided a total of $n = 2,299$ surveys.”

6. How many completed days (7.23 out of 8) were days with cortisol sampling?

Response: I have now added this information to the manuscript on p.10: “Participants completed 7.23 out of 8 scheduled surveys ($SD = 1.66$) and provided 3.08 ($SD = 1.02$) out of 4 days of valid cortisol data, on average.”

7. I appreciate the inclusion of the statistical power for the main analyses, but given the cortisol analyses are with 255 participants, did the author run the same power analysis with the slightly smaller sample size?

Response: Thank you for this comment. For the diurnal cortisol analyses with $N = 255$ participants, power was smaller than for the larger sample size of $N = 318$ participants (power of $> 90\%$), providing approximately 71% power to detect small within-person effects and 61% power to detect moderate cross-level interactions. This information is now added to

the manuscript on p. 15: “For diurnal cortisol data, a sample of N = 255 participants provided 71% power to detect small within-person effects and 61% power to detect moderate cross-level interactions.”

8. I am surprised that the participants reported positive solitude so often - could this be because they, on average, have older children (and have had 7 or so years to adjust to daily schedules)?

Response: There was a small to moderate significant correlation between average personal time and age of youngest child in the dataset ($r = 0.18, p < .001$). The revised discussion now addresses the fairly frequent occurrence of personal time, incorporating the reviewer's thoughts:

(p. 20): “Participants reported that they had the opportunity to take time for themselves on 4 out of 5 days, on average. Parents might have learned over time to prioritize personal time, recognizing its importance for well-being, or have developed strategies to integrate restorative activities into their daily routines since having their first child.”

9. In the results section, it is unclear why the author is reporting on daily stressor exposure. Although interesting, it is not a hypothesized pathway; justification in the introduction should be added, or this should be dropped.

Response: When selecting covariates, I included study day and stress exposure in the main models, while other variables (e.g., number of children in the household, age of the youngest child) were tested in sensitivity analyses. Daily stress exposure is a crucial covariate because it is related to both affect and cortisol and may indicate a busier schedule that reduces opportunities for personal time. Controlling for stress exposure allows me to examine the unique contribution of personal time to these outcomes, holding stress constant. As the reviewer suggested, this rationale has now been added to the introduction:

(p. 9): “Daily stress exposure was controlled for to account for its potential confounding influence on affect and cortisol^{35,38}, as busier, more stressful days may limit opportunities for personal time. This allows the unique association of personal time with daily well-being to be examined, holding daily stress exposure constant.”

10. Out of curiosity, is it possible to determine how often solitude and leisure time co-occurred? In the discussion the author mentioned solitude and self-care activities that were similar to leisure activities, so it would be curious to know.

Response: In response to the reviewer's comment, I examined the co-occurrence of time to oneself and leisure time. On days when participants reported having had personal time, they spent, on average, about one hour more in leisure activities (days with personal time: 3.28 hrs of leisure; days without personal time: 2.29 hrs of leisure). Using a binary indicator to distinguish days with no leisure versus at least some leisure, the two measures were significantly associated (see figure below): days with at least some leisure and days with at least some personal time co-occurred on 1,696 of 2,299 measured occasions (73.8% of days). This information has now been added to the sensitivity and follow-up analyses.

		timeoneself			
leisureyesno	0	1	Total		
0	141 (55.3%)	114 (44.7%)	255 (100.0%)		
1	346 (16.9%)	1696 (83.1%)	2042 (100.0%)		
<NA>	0 (0.0%)	2 (100.0%)	2 (100.0%)		
Total	487 (21.2%)	1812 (78.8%)	2299 (100.0%)		

$$\chi^2 = 197.2736 \quad df = 1 \quad p = .0000$$

(p. 18-19): “Participants also reported the number of hours spent on leisure activities each day. Days with personal time were associated with more leisure, with participants spending an average of 3.28 hours in leisure on personal time days versus 2.29 hours on days without personal time. Using a binary indicator, days with personal time co-occurred with days with at least some leisure on 73.8% of measured occasions. When leisure time was used in place of personal time as the predictor, the same pattern of results was observed: On days when participants spent more time on leisure activities than usual, they reported higher positive affect ($\beta = 0.05$, $SE = 0.01$, $p < .001$) and lower negative affect ($\beta = -0.04$, $SE = 0.02$, $p = .031$), holding stress exposure constant. Leisure time was also significantly associated with cortisol slope—days with more time spent on leisure were linked to a steeper cortisol decline ($\beta = -0.08$, $SE = 0.04$, $p = .034$).”

(p. 21): “In the current study, personal time often coincided with leisure activities, with participants engaging in roughly an hour more leisure on days when they had the opportunity for personal time. For parents, using moments of personal time for self-care—such as napping, listening to soothing music, or engaging in calming and restorative activities like painting or physical exercise—may offer important opportunities for psychological disengagement and emotional recovery from daily pressures³³.”

Reviewer 2's comments:

Thank you for the opportunity to review this paper. Overall, I think this is an interesting study, and I want to commend the researcher for sharing their code and for explaining the analyses and results very clearly. The results are also presented in a way that is easy to follow. That said, my main concern is with the framing of the research.

Response: I thank the reviewer for their positive words and address their concerns point-by-point below.

*1. I am not convinced the study is really about solitude. The measure used—"Since this time yesterday, did you have the opportunity to take time for yourself?"—does not necessarily capture solitude. Prior work, including Nguyen et al. (2025)'s work on mothers, shows that "personal time" can include self-care or enjoyable activities, sometimes even bonding with one's baby. In other words, "time for yourself" does not always mean being alone. This is why I think the paper should be framed around "time for yourself" (or "me-time") rather than solitude. Solitude may be part of it, but the construct as measured here is broader. Also, here is the latest reference! Nguyen, T. T., Konu, D., Tetteh, D., Tshimbalanga, P., Weissová, J., & Xiong, M. (2025). "I got all sorts of solitude, but that solitude wasn't mine": A Mixed-Methods Approach to Understanding Aloneness during Becoming a Mother. *Journal of British Psychology*. Doi: <https://doi.org/10.1111/bjop.70019>*

Response: Reviewer 1 had a similar concern pertaining to conceptual clarity. In line with your suggestion, I have now framed the manuscript around "personal time" and "time for oneself" rather than solitude. In particular, the revised manuscript now refers to more literature on the well-being implications of personal and leisure time in the introduction. I have also updated the reference of Nguyen et al., 2025, to the published form of the preprint article.

2. I suggest restructuring the Introduction. It would work better if it started with the literature on parents and the complexity of their solitude, then discussed research on solitude while still emphasized that, for parents, it is about "me-time", and finally personality differences. As pointed out in #1, it is important that you distinguish between solitude and "me-time". This distinction should be made clear in the Introduction. Otherwise, there's a risk of muddling the solitude literature.

Response: I very much agree with the reviewer. The previous manuscript started out by describing the importance of solitude. Instead, the first paragraph now focuses on the importance of personal time for parenting:

(p. 3): "Many adults report feeling short on personal or leisure time^{1,2}. A subjective feeling of lack of personal time has been associated with diminished well-being and lower quality of life^{3,4}. Time strains are particularly pronounced for midlife parents, who are often juggling multiple roles at once—managing family responsibilities, caring for aging parents, and working in high-demand career stages^{5,6}. This study explores the role of personal time for the daily emotional and physiological well-being of parents of underage children, and examines whether these associations are moderated by personality."

From there, I outline the importance and benefits of personal time, making sure to conceptually distinguish these concepts from solitude:

(p. 3): “Personal time refers to time that an individual spends in self-directed, restorative or voluntary activities, free from obligatory work, caregiving, or household duties ^{7,8}. Personal time can but does not necessarily need to happen when alone, defined as the physical absence of other people, or in solitude, defined as the lack of social interaction ⁹.”

(pp. 3-4): “In the life-balance literature, research indicates that a three-dimensional model—including work time, social time, and personal time—better predicts health outcomes than a two-dimensional model based only on work and social time ¹³. Research has linked engaging in leisure activities or activities that are driven by intrinsic motivation with enhanced recovery, defined as the process to replenish resources like energy, attention, and mood when not exposed to further demands ^{21,22}”

3. *Similarly, I don't think the item fully captures positive solitude (which is solely about solitude) as it has been conceptualized in works like Weinstein et al. or Ost-Mor et al. “Me-time” overlaps with positive solitude but for parents, can capture bonding time with baby or social time when someone else take over childcare to leave parents some space. See Nguyen et al. (2025)'s Discussion section.*

Response: Thanks for this important comment. I now discuss that personal time does not need to occur in solitude in the revised discussion, referring to Nguyen et al. (2025):

(p. 22): “It is also important to recognize that while some personal time may happen in solitude, many parents experience time spent with their children as deeply fulfilling and restorative. In young mothers, time for oneself does not necessarily involve physical separation from the baby; instead, it can coincide with moments of rest and relaxation ³³. In a study of 186 parents of young children, participants tracked their daily activities and rated how much they enjoyed each one⁷³. Interestingly, time spent on childcare was linked to greater positive affect than any other activity, including watching TV or cooking.”

4. *The Discussion should also be revised to reflect this reframing. Right now, equating “time for yourself” with positive solitude is problematic, especially without confirmation that participants who said yes were in fact alone and not interacting with others.*

Response: All throughout the discussion, I now use the terms “personal time” or “time to oneself” rather than positive solitude when referring to the current study.

5. *In addition, I would like to request some clarification about the sample and descriptive data. Specifically, could the authors provide more information on how healthy or stressed this sample of parents was? This would help clarify, especially because I wonder if the sample may have more support, given that 79% of days were coded as “yes” for having time to themselves. Since there are relatively fewer data points for “no” time to themselves, how does this imbalance affect the generalizability of the results? Overall, I would like to know a*

bit more about this group of parents and how representative they might be. I think this paper addresses an important and timely issue. The strengths are the open sharing of code, the clear explanation of analyses, and the accessible presentation of results. With reframing and some additional detail on the sample, this paper could make a strong contribution.

Response: As the reviewer requested, I have added the following information about participants' stress and health to the manuscript:

(pp. 10-11): "Participants had an average age of 40.06 years ($SD = 7.54$), 55% were women, and the average annual household income was \$97,435.71 ($SD = \$65,183.49$). They lived with an average of 2 underage children ($M = 2.02$, $SD = 1.15$) and the youngest child was 7.61 years old, on average ($SD = 5.19$). The majority of the sample identified as White (86.5%), 5.0% as Black, 1.3% as native American, 0.9% as Asian, and 6.3% as other. Participants rated their subjective health as fairly good ($M = 7.52$, $SD = 1.48$) on a 0–10 scale, where 0 indicated "the worst possible health" and 10 indicated "the best possible health." Participants reported experiencing any stressor on approximately half of the days ($M = 0.47$, $SD = 0.28$). The most commonly reported stressor was avoiding a disagreement ($M = 1.37$ out of 8, $SD = 1.33$), followed by an argument or disagreement ($M = 1.03$, $SD = 1.19$), a stressor at work or school ($M = 0.93$, $SD = 1.13$), a stressful event at home ($M = 0.65$, $SD = 1.02$), a stressful event that happened to a close other ($M = 0.30$, $SD = 0.62$), and discrimination ($M = 0.02$, $SD = 0.12$)."

In multilevel models, 21% of days coded as having no personal time still allows for reliable estimation due to the large number of level-1 observations (483 out of 2,299). However, biased parameter estimates can happen when many individuals show little or no variability across days for a given predictor (i.e., no variation in the level-1 variable of interest; see Ram et al., 2021). In our sample, only three individuals never had any time to themselves across the eight days (< 1% of the sample), posing no threat to statistical validity.

Reference:

Ram, N., Brinberg, M., Pincus, A. L., & Conroy, D. E. (2017). The questionable ecological validity of ecological momentary assessment: Considerations for design and analysis. *Research in Human Development*, 14(3), 253–270.
<https://doi.org/10.1080/15427609.2017.1340052>

Reviewer 3's comments:

This study examined how parent solitude is associated with parent negative affect and cortisol (an indicator of stress). Overall the study is well-written, and the topic is interesting and well-developed. More information is needed throughout the differentiate terms, and important methodological information and additional analyses should be added. Suggestions are also given regarding streamlining the introduction to focus more specifically on study findings and not extend beyond the scope of this paper.

Response: Thank you for your positive feedback on the manuscript. I have addressed all comments below and revised the manuscript accordingly.

1. *Abstract: Add child ages to the abstract.*

Response: As suggested, I have added the descriptives of age of the youngest child to the abstract (p. 2): “A sample of 318 parents (Mage = 40.06 years, $SD = 7.54$; 45% male) with underage children (Mage of youngest child = 7.61 years, $SD = 5.19$) completed up to 8 consecutive days ...”

2. *It would strengthen the paper to have clearer definitions throughout, as several of the studies reviewed looked at solitude but some at related concepts (such as time alone). For example, the introduction states that solitude is different than being alone, yet throughout the paper discussion of solitude refers to the benefits of time alone.*

Response: Reviewers 1 and 2 had similar concerns with respect to conceptual clarity (see Reviewer 1’s comment 2 and Reviewer 2’s comment 1), given that the measure used asked “*Since this time yesterday, did you have the opportunity to take time for yourself?*”. They suggested to frame the manuscript around “personal time” and “time for oneself” rather than solitude. While solitude may be one aspect of personal time, the construct in this study encompasses a broader category. I have therefore revised the manuscript (particularly referring to more literature on the well-being implications of personal and leisure time in the introduction) to reflect this change. I also made sure to include clear definitions in the introduction, e.g.:

(p. 3): “Personal time refers to time that an individual spends in self-directed, restorative or voluntary activities, free from obligatory work, caregiving, or household duties^{7,8}. Personal time can but does not necessarily need to happen when alone, defined as the physical absence of other people, or in solitude, defined as the lack of social interaction⁹.”

3. *More is also needed to understand how solitude relates to privacy (and how privacy is defined).*

Response: As recommended, the revised manuscript now defines privacy more clearly and strengthens the conceptual link between privacy and personal time:

(p. 3): “Early research on personal time emphasized privacy—the selective control of access to the self¹⁰— as a way individuals control interactions with others, uphold personal boundaries to preserve self-identity, and regulate well-being. Pedersen¹¹ identified five core functions of privacy that may be facilitated by personal time: autonomy (independence, freedom from social pressure), confiding (intimacy, expressing true emotions), rejuvenation (taking refuge, recover), contemplation (self-reflection), and creativity (problem-solving, idea generation).”

4. *How much does solitude matter vs what you do during solitude? It seems important to make this distinction throughout the introduction, as studies reviewed look at multiple related concepts, and clearly distinguishing would enhance the paper.*

Response: Thank you for this helpful suggestion. In the revised manuscript, I now explicitly frame the central construct as **personal time**, defined as time spent free from external demands and available for self-directed activities. I distinguish this construct from related concepts—including *solitude*, *leisure time*, and *privacy*—while acknowledging their conceptual overlap. Throughout the introduction, I now clarify how the present work builds on but is distinct from these prior literatures, emphasizing that the current study investigates not merely the state of being alone but the opportunity to take time for oneself. This distinction is now consistently maintained across the manuscript.

5. It would also enhance the paper to discuss the implications of these distinctions for understanding the strengths/gaps of the literature as a whole. Are there other ESM studies that have examined solitude? What do we know from prior work about how much it varies from day to day? And if it is a trait?

Response: I agree that discussing these distinctions can clarify the strengths and gaps in the literature. In the revised manuscript, I now highlight that while prior experience sampling studies have examined *solitude*, most have not distinguished between mere time spent alone and the quality or type of activities engaged in during that time. Evidence from prior work suggests that personal time—and related constructs like solitude or leisure—varies substantially from day to day, reflecting both situational factors (e.g., work or caregiving demands) and individual differences. I have incorporated these points to better contextualize the current study within the broader literature.

(pp. 26-27): “Measurement limitations should also be noted. Personal time was assessed using a single binary item asking whether participants had time to themselves on a given day (yes/no), which may oversimplify the complexity of experiences of personal time. Evidence from prior experience sampling studies suggests that personal time and solitude fluctuate substantially from day to day in response to situational demands while also exhibiting modest trait-like stability across individuals^{8,87}. Future research should employ more nuanced measures that capture the different types and qualities of personal time, as well as its interplay with related concepts. While similar patterns were observed when using hours spent on leisure activities as a predictor, the daily interplay between personal time and related constructs such as solitude, privacy, and leisure remains to be explored. This study did not assess what specific activities parents engaged in during their time to themselves, making it difficult to determine whether benefits are driven by personal time per se or by the activities undertaken during that time. For instance, whether personal time involves technology use, physical separation from others, or internal focus may significantly influence whether personal time is experienced as restorative⁹. Moreover, personal time used for passive entertainment like watching Netflix or social media, recreational activities like going for a run, contemplative activities like planning or reflecting, or not doing any activity at all (activity-less personal time) may have differing effects on daily well-being⁴⁸.”

6. Dysregulated cortisol can also lead to cortisol hyperarousal, and this should be discussed as well in the introduction. .

Response: As suggested, the revised introduction now includes this important point:

(p. 7): “Even single- or few-day measures of flattened diurnal cortisol slopes are linked to negative health effects, suggesting that sustained reductions in physiological recovery may gradually lead to maladaptive HPA axis regulation⁴⁰. HPA axis dysregulation, in turn, can result in abnormal total cortisol output, either excessive (hypersecretion) or diminished (hyopsecretion)⁴².”

7.It would be helpful to add more about burnout when you are discussing the experiences of parents and stress, especially as you cite burnout literature when discussing personality. These are distinct concepts and the nuances of their differences should be made clear when reviewing the literature.

Response: Thanks for this comment. I have now added a definition of parental burnout to the revised manuscript:

(p. 5): “Thus, not surprisingly, parental stress is robustly linked to reduced well-being³⁰ and can lead to parental burnout, a condition characterized by chronic exhaustion related to parenting, emotional distancing from one’s children, and a sense of being an ineffective parent³¹.”

8.More explicit hypotheses on specific personality traits would enhance the paper.

Response: I am hesitant to specify explicit hypotheses regarding personality traits because the existing literature on personality differences in solitude and well-being is highly inconsistent—for example, findings regarding introversion are divergent. Even less is known about how personality relates to benefits of personal time specifically, leaving an insufficient evidence base to formulate clear, trait-specific predictions.

9.Check reference formatting (minor).

Response: Thank you for the careful review of the manuscript. I have thoroughly checked all references and corrected any formatting errors.

10.Much more information is needed in the methods section. How often and when were individuals in the refresher sample surveyed? Does the refresher sample differ from the full sample? How was the daily diary data collected? What time of day was it collected?

Response: Individuals in the refresher sample were surveyed only once, to replenish and extend the original MIDUS cohort. Participants in the daily diary study (conducted between 2012 and 2016) were randomly selected from the main survey participants. Data was collected via evening telephone surveys (the exact time of interview is not denoted in the data). Sociodemographic characteristics of the participants in the daily diary study were similar to those of participants in the main survey.

Table 1 in Surachman et al., (2018). <https://doi.org/10.1093/geronb/gby014>.

Demographic variable	MIDUS Refresher baseline survey^a	MIDUS Refresher daily diary study^b
Age		
Young adults (23–39)	27.3	27.7
Midlife adults (40–59)	39.9	51.2
Older adults (60–76)	32.8	21.1
Gender (%)		
Males	48.1	44.4
Females	51.9	55.6
Education (%)		
No bachelor degree	50.1	53.9
Bachelor degree and above	49.9	46.1
Average household income in USD (SD)	85,285.55 (64,367.35)	84,458.33 (67,206.75)
Marital status (%)		
Married	64.0	65.3
All others	36.0	34.7
Children in household^c (%)		
Yes	45.1	45.8
No	54.9	54.2
Race (%)		
Caucasian	82.3	82.6
African American	7.7	7.4
All other races	10.0	10.0

Note: MIDUS Refresher = Midlife in the United States Refresher Study.

^aRespondents in the MIDUS Refresher baseline survey ($N = 3,577$). ^bRespondents in the MIDUS Refresher daily diary study, all of whom had previously participated in the MIDUS Refresher main survey ($N = 782$). ^cWhether respondent had at least one child aged 18 or younger living in the house.

In response to the reviewer's comment, I have added this information to the manuscript:

(pp 9-10): "Between 2011 and 2014, an additional sample (Refresher Cohort) of 3,577 adults aged 25 to 74 was recruited to replenish the number of middle-aged adults in the original MIDUS cohort. Participants were recruited through random digit dialing. The MIDUS Refresher survey used the same assessments as the original study, where participants first completed 30 min baseline phone interviews followed by self-administered questionnaires via mail. A random subsample of participants ($N = 782$) was selected to take part in a Daily Diary project. On 8 consecutive evenings, participants completed phone surveys about their experiences over the past day. These daily surveys covered various aspects of their daily lives, including stressors, emotions, physical symptoms, and social interactions. The sociodemographic

characteristics of participants in the Daily Diary study closely resembled those of the broader Refresher survey sample⁵³.”

11. The measure of positive solitude should be described more in relation to the literature. From the question asked in the survey, this seems to get more about being alone than necessarily solitude. And there is nothing in the question as I read it, that gets at whether or not that solitude is positive or negative. It seems to get at time alone only.

Response: As noted above, I now use the term **personal time** to clearly differentiate this construct from related concepts such as solitude, leisure, and privacy. The survey item—“Since this time yesterday, did you have *the opportunity to take time for yourself?*”—captures whether participants had time to themselves for self-directed, restorative activities, reflecting a construct with a positive valence.

12. More information about cortisol measures and analysis should be added here. Specifically, more information is needed on collection and processing should be added here (rather than referring to another paper only). Was the data log transformed- and if so how? How many people provided full cortisol samples and both samples needed to calculate the score for this paper? Did people dropped for noncompliance differ from those that were compliant? Why did you divide your score of the slope by the time elapsed between the samples?

Response: Thank you for this helpful suggestion. I have now expanded the description of cortisol collection and processing procedures in the manuscript. Specifically, I added information on the number of participants who provided valid cortisol samples and clarified the inclusion criteria (i.e., at least one valid day with both waking and bedtime samples). I now report that 14 individuals were excluded for non-compliance, and provide a comparison of those excluded for non-compliance with those retained in the analyses. I have also clarified that cortisol values were log₁₀-transformed and described the computation of the diurnal slope, including the rationale for dividing by the time elapsed between samples—namely, to account for variation in total time awake and align with best practices for estimating diurnal cortisol slopes. I further explained why the wake–evening slope is used as a robust indicator of diurnal cortisol decline. These details are now included in the revised methods section.

(pp. 12-13): “Cortisol data were available from 269 individuals. Fourteen individuals were excluded because they did not provide at least one valid day of cortisol data (i.e., a waking and a bedtime sample). Thus, the final analytic sample consisted of 788 valid cortisol days, completed by 255 participants. Those excluded due to non-compliance did not differ from included participants in terms of sex, ethnicity, age, subjective health, average personal time, or positive or negative affect, but they did report significantly lower income ($M = 77,160.71$ vs $98,380.83$, $t(18.93) = 2.30$, $p = .033$). On average, participants provided 3.08 days ($SD = 1.02$) of cortisol data. Cortisol values were log₁₀-transformed. For each day, the diurnal slope was calculated by subtracting the morning from the evening cortisol value, dividing by the number of hours between the two samples⁵⁹. Dividing by time elapsed between samples accounts for differences in total time awake—reflecting the duration over which cortisol can decline—and is the recommended method for estimating diurnal cortisol slopes⁴². The

wake–evening cortisol slope is the most commonly used marker of diurnal cortisol decline and less vulnerable to distortion from daytime outliers, as compared to regression-based estimation methods that use all daily samples⁴¹.”

13. It would strengthen the paper to have a data analysis plan. I had many questions as I was reading the methods section. Some of them were answered later (and some not). It would enhance the paper to discuss your steps in detail earlier in the paper. For example, more information is needed earlier about the other control variables that you also tested, but then later dropped (age, sex, etc.). If you planned to test the moderators separately or together, etc.

Response: As suggested, the revised statistical analysis section now provides a more detailed description of the data analysis plan. In line with your earlier comment, it also specifies the level at which each variable was included:

(pp. 14-15): “A first set of models predicted daily affect and diurnal cortisol slopes by study day, daily personal time, and stressor exposure at level 1 and by person-mean personal time, person-mean stressor exposure, and the five personality traits at level 2 (main effects only model; H1). In a second step, all five interaction terms between daily personal time and personality traits were entered simultaneously into the models at level 2 (moderation model; H2). Sensitivity analyses tested whether findings remained consistent when including all the following additional covariates into the models: age, income, number of children, age of youngest child (all models); medication intake, subjective health (models for cortisol).”

14. Did you control for medications known specifically to interfere with cortisol?

Response: In the sensitivity analyses reported in the paper, I previously controlled for whether participants had taken any medication in the past 30 days (1 = yes, 0 = no). In response to the reviewer’s comment, I reviewed the codebook and confirmed that the survey also specifically asked whether participants took medications known to influence cortisol on the days they provided saliva (including steroid nasal sprays, steroid inhalers, oral steroids, cortisone creams, corticosteroid injections, antidepressants or anti-anxiety medications, or other hormonal medications). This applied to 30% of the sample (77 out of 255 participants with cortisol data). Although this variable was not significantly associated with the diurnal cortisol slope, I updated the control variable to this more targeted measure, as the reviewer noted that controlling for medications that directly interact with cortisol is conceptually more appropriate.

15. How many individuals were dropped due to having too much missing data for the personality measures?

Response: One individual was missing data for personality, this is now mentioned in the methods section (p. 10): “One person was missing information on personality, resulting in a sample size of N = 318 participants who provided a total of n = 2,299 surveys.”

16. Were the personality measures added as level two predictors only?

Response: Yes, personality was assessed only once as a trait characteristic. Thus, personality was included as a level-2 predictor at the person level, only. This is now mentioned in the data analysis plan, which was added in response to your earlier comment:

(p. 14-15): “A first set of models predicted daily affect and diurnal cortisol slopes by study day, daily personal time, and stressor exposure at level 1 and by person-mean personal time, person-mean stressor exposure, and the five personality traits at level 2 (main effects only model; H1). In a second step, all five interaction terms between daily personal time and personality traits were entered simultaneously into the models at level 2 (moderation model; H2).”

17. What was the racial makeup of your sample (beyond what % was white)?

Response: Reviewer 2 also wanted additional information about sample descriptives. We now provide a more elaborate overview of the sample, including the racial makeup:

(pp. 10-11): “Participants had an average age of 40.06 years ($SD = 7.54$), 55% were women, and the average annual household income was \$97,435.71 ($SD = \$65,183.49$). They lived with an average of 2 underage children ($M = 2.02$, $SD = 1.15$) and the youngest child was 7.61 years old, on average ($SD = 5.19$). The majority of the sample identified as White (86.5%), 5.0% as Black, 1.3% as native American, 0.9% as Asian, and 6.3% as other. Participants rated their subjective health as fairly good ($M = 7.52$, $SD = 1.48$) on a 0–10 scale, where 0 indicated “the worst possible health” and 10 indicated “the best possible health.” Participants reported experiencing any stressor on approximately half of the days ($M = 0.47$, $SD = 0.28$). The most commonly reported stressor was avoiding a disagreement ($M = 1.37$ out of 8, $SD = 1.33$), followed by an argument or disagreement ($M = 1.03$, $SD = 1.19$), a stressor at work or school ($M = 0.93$, $SD = 1.13$), a stressful event at home ($M = 0.65$, $SD = 1.02$), a stressful event that happened to a close other ($M = 0.30$, $SD = 0.62$), and discrimination ($M = 0.02$, $SD = 0.12$).”

18. Table 2 would be strengthened by labeling the within and between person effects more clearly

It is unclear if the moderators were tested individually? And while controlling for the other personality traits? Or if all the moderators were tested in the same model? And did the authors also test moderation for the between-person effects?

Response: As suggested, I have reordered and clearly labeled the level-1 fixed effects, level-2 fixed effects, and random effects in Table 2. The revised statistical analysis section also clarifies that the moderating effects of all five personality traits were tested simultaneously in the same model as cross-level interactions (see my response to your earlier comment).

Because this study focused on day-to-day fluctuations rather than the between-person effects of person-mean personal time on average affect and cortisol slope, I did not examine moderation of these between-person effects by personality in the paper. In response to your comment, I ran those models, and no significant moderation effects were observed.

19. *The sensitivity and follow up analyses are interesting, and some of it would enhance the paper and I recommend moving some of it and expanding it throughout. For example, given that solitude's effects may be solitude and what you do with that time, adding leisure time would enhance the paper. Adding the lagged models to the main body of the paper would also strengthen it. The question on daily stressors as a potential moderator seems less relevant to the overall question of this paper, and I recommend dropping it.*

Response: As suggested by the reviewer, I have now added the findings on lagged analysis to the main body of the results and have removed the follow-up analysis pertaining to daily stressors as a potential moderator.

I hesitate to include leisure time in the main analyses because my hypotheses specifically focused on personal time, defined as self-directed, restorative activities free from obligatory work, caregiving, or household duties (see p.3 in the revised introduction; in line with the reframing of the paper, as mentioned above). While leisure can include hobbies, socializing, or entertainment, not all of these activities qualify as personal time—for example, attending a required family event or participating in a group activity out of obligation may be leisure but not genuinely restorative.

Thus, I was particularly interested in the question, “**Did you have the opportunity to take time for yourself?**” in the MIDUS refresher dataset, because it captures the sense of autonomy and self-direction. I expected this type of personal time to be more strongly related to affect and health than general leisure, as might provide psychological respite, enhance feelings of control, and buffer the impact of daily stressors. The revised introduction now more clearly outlines mechanisms through which personal time might link to improved daily health and affective well-being (pp. 3-4). However, if the reviewer and the editor strongly feel that leisure should be included in the main results, I am happy to do so.

20. *The discussion would benefit from being more streamlined around the main questions of this paper.*

The discussion goes beyond the main analysis of this paper (in that it discusses what individuals do with solitude, which was not assessed here). However, adding the leisure findings more thoroughly would enhance this, and expand the ability of authors to discuss this in the discussion. Also, the authors discuss potential downsides of solitude, but findings do not support this- so be sure to discuss those in context specifically of what you find. And how your findings support other work or inspire future research questions based on gaps in information that you had or contradictions with other studies.

Response: Thank you for this important comment. Reviewer 1 also commented on the conceptual overlap between personal time and leisure activities (see their comment 10). In response to both of your comments, I examined the co-occurrence of time for oneself and leisure time. On days when participants reported having had personal time, they spent, on average, about one hour more in leisure activities (days with personal time: 3.28 hrs of leisure; days without personal time: 2.29 hrs of leisure). Using a binary indicator to distinguish days with no leisure versus at least some leisure, the two measures were significantly associated ($\chi^2 = 197.27, p < .001$): days with at least some leisure and days with at least some personal time co-occurred on 1,696 of 2,299 measured occasions (73.8% of days). This information has now been added to the follow-up analyses.

(p. 18-19): “Participants also reported the number of hours spent on leisure activities each day. Days with personal time were associated with more leisure, with participants spending an average of 3.28 hours in leisure on personal time days versus 2.29 hours on days without personal time. Using a binary indicator, days with personal time co-occurred with days with at least some leisure on 73.8% of measured occasions. When leisure time was used in place of personal time as the predictor, the same pattern of results was observed: On days when participants spent more time on leisure activities than usual, they reported higher positive affect ($\beta = 0.05$, $SE = 0.01$, $p < .001$) and lower negative affect ($\beta = -0.04$, $SE = 0.02$, $p = .031$), holding stress exposure constant. Leisure time was also significantly associated with cortisol slope—days with more time spent on leisure were linked to a steeper cortisol decline ($\beta = -0.08$, $SE = 0.04$, $p = .034$).”

In response to your comment, I have considerably reworked the discussion, to now more clearly focus on the importance of personal time for parenting. For example, the section that discussed potential downsides of solitude has been removed.

(p. 20): “Among the various restorative functions of personal time, previous literature has pointed out its role in emotion regulation²². Individuals tend to seek time for oneself particularly when experiencing high-arousal negative emotions like irritation or anxiety⁶⁴. In this study, parents reported higher levels of positive emotions—such as happy, calm, cheerful, and satisfied—and lower levels of negative emotions—such as anger, fear, frustration, and sadness—on days when they had some time for themselves. This is in line with other daily diary research linking personal time to higher vitality, increased positive affect, and decreased negative affect^{8,65}.”

(p. 21): “In the current study, personal time often coincided with leisure activities, with participants engaging in roughly an hour more leisure on days when they had the opportunity for personal time. For parents, using moments of personal time for self-care—such as napping, listening to soothing music, or engaging in calming and restorative activities like painting or physical exercise—may offer important opportunities for psychological disengagement and emotional recovery from daily pressures³⁵.”

In addition, I have added more information on how findings could guide future research:

(p. 20-21): “Thus, it would be interesting for future research to investigate how personal time facilitates emotional down-regulation in different affective contexts, and whether its restorative value differs across life stages (e.g., early parenthood vs. caregiving for aging parents). Related research with middle-aged and older informal caregivers suggests that time for oneself can foster self-connection and enhance well-being—particularly when approached with a mindset of self-kindness⁶⁸. Another important question concerns the dose–response relationship of personal time and well-being—that is, how much is needed, and at what times, to be beneficial? Bradshaw et al.⁶⁶ found that even brief periods of time to oneself—as short as five minutes—led to mood improvements and reduced feelings of depletion. Future research should examine whether personal time earlier in the day influences subsequent stress

reactivity and emotional recovery. It will also be important to investigate how patterns of personal time across multiple days accumulate and contribute to longer-term health and well-being.”

(p. 26): “Future research should employ more nuanced measures that capture the different types and qualities of personal time, as well as its interplay with related concepts. While similar patterns were observed when using hours spent on leisure activities as a predictor, the daily interplay between personal time and related constructs such as solitude, privacy, and leisure remains to be explored. ”

(p. 27): “This study did not assess what specific activities parents engaged in during their time to themselves, making it difficult to determine whether benefits are driven by personal time per se or by the activities undertaken during that time. For instance, whether personal time involves technology use, physical separation from others, or internal focus may significantly influence whether personal time is experienced as restorative⁹. Moreover, personal time used for passive entertainment like watching Netflix or social media, recreational activities like going for a run, contemplative activities like planning or reflecting, or not doing any activity at all (activity-less personal time) may have differing effects on daily well-being⁴⁹.”

22. Similarly it would strengthen the paper to streamline the discussion of personality differences, and focus it more on the questions asked here and how these are/are not supportive of other literature and possible explanations and things you could not answer but would be next steps.

The limitations section was very thoughtful and thorough.

Response: Thank you for this helpful suggestion. I have revised the discussion of the personality moderation findings to focus more clearly on how the results align with existing literature, what they suggest theoretically, and what remains unanswered. I also added concrete next steps to address the underlying mechanisms in future research. Examples are:

(pp/ 22-23): “The decrease in negative affect on days with personal time was greater for men than for women high in neuroticism. This aligns with prior research indicating that the link between feelings of time strains for oneself and life satisfaction was stronger for fathers than for mothers⁵. Women’s leisure time might be of lower quality than men’s because it tends to be more limited, frequently interrupted by caregiving demands, and squeezed into busy schedules rather than planned which could reduce its restorative value⁷⁴. Future research should examine how the context and quality of personal time, not only its quantity, interact with personality to predict daily emotional recovery across genders.”

(p. 23): “Because this study did not include direct measures of momentary physiological arousal or coping responses, it remains unclear whether the benefits of personal time for parents high in neuroticism stem from reduced stress reactivity, cognitive distance, emotional recovery, or simply relief from external demands^{45,74}. Future work could strengthen this understanding by incorporating more detailed daily assessments of coping processes, stress reactivity, and additional indices of cortisol regulation (e.g., total cortisol

output via area-under-the-curve) to better capture how personal time supports those high in neuroticism.”

(p. 23-24): “Openness also moderated the relationship between personal time and negative affect, such that individuals with higher levels of openness experienced a greater reduction in negative affect on days when they had time for themselves. This aligns with existing research linking openness to curiosity, imagination, arts, and intrinsic engagement in self-directed novel activities ⁵⁰. People high in openness have been found to value personal time as a means for creativity and self-exploration ⁵². These findings suggest that personal time may allow highly open parents to engage in personally meaningful or imaginative activities such as journaling, creative writing, painting, listening to music, or reading fiction ⁷⁷. To better understand why personal time is especially beneficial for parents high in openness, future studies could capture concepts such as creativity, absorption and, flow experiences ⁷⁶.”

Reviewer 1's comments:

I commend the author for the massive and thorough rebuttal of all reviewer comments. For only one author, this can be quite an undertaking. I have a handful of points based on the revised manuscript; however, all are minor.

Response: I thank the reviewer for highlighting the strengths of the revised paper and address all additional minor comments below.

1. I appreciated the reminder to reviewer 2's point that 21% of days with no personal time still allows for an examination of the questions at hand. I wonder if adding two things would strengthen this point. Notably, with an eight-day diary study this would mean that ~3 days were no personal time (on average) and ~5 were with personal time. Although unbalanced, that still provides plenty of days with and without personal time. The other piece could be to clarify (perhaps for early career readers without the statistical understanding) that you ran unconditional/empty models to identify the between- and within-persons variance is across the sample for your variables (i.e., page 16) which indicated that 75% of the variance of personal time, for example, is at the day-to-day level (or within-persons).

Response: I have added these details that clarify the statistical approach to the revised results section:

(p. 15): "Participants reported personal time on 79% of days ($SD = 0.25$), averaging about 2 days out of every 8 days without personal time."

(p. 16): "We ran unconditional models to identify variance composition at the day and person level. All four daily assessed parameters showed significant variation on a day-to-day level."

2. Thank you for the additional analyses throughout to answer our questions/comments. I am unsurprised that leisure time and time alone are associated with one another - and I agree that especially for parents, alone time may be the only time that they can engage in leisure activities.

Response: Thanks, I have added an additional sentence to the conclusion of the revised manuscript to highlight this important point:

(p. 28): "For parents, periods of personal time may constitute their primary—if not only—opportunity to engage in leisure pursuits, such as reading for pleasure, exercising, engaging in creative hobbies, or simply resting without caregiving demands."

3. Could you clarify how the Refresher daily diary project had an N of 782 but the final analytic sample was N = 318? Based on what's written, it seems like only 318 people had an underage child living in the household, but it wasn't entirely clear.

Response: Indeed, of the 782 participants, 319 reported having at least one underage child living in the household. One participant was missing personality data. The revised manuscript now clarifies how the final analytic sample was determined:

“Out of the 782 participants, 319 individuals had at least one underage child living in their household (biological child, adopted child, stepchild, or child of partner). One person was missing information on personality, resulting in a sample size of N = 318 participants who provided a total of n = 2,299 surveys.”

Reviewer 2's comments:

Thank you for your thorough revision. It is great that you have reframed the work so it more aligns with the construct "time for oneself" that you measured. I also thought it was nice to see more statistics around what those "yes" response to having time for oneself means. Your revision has satisfied my concerns.

Response: Thank you for your thoughtful feedback. I am pleased that the revisions have addressed your concerns.

Reviewer 3's comments:

The authors have addressed most of my concerns. I appreciate the improved conceptual clarity and thoroughness of this paper and expect it will make an important contribution to the field. I have a few minor suggestions for further revision.

Response: Thank you for your positive feedback on the revised manuscript.

1. I appreciate the focus on personal time and conceptual clarity. Given your measure captures opportunities for personal time (as opposed to whether they actually had personal time)- I would recommend capturing this nuance throughout. For example by specifying opportunities for personal time in the abstract, results, discussion, limitation.

Response: As suggested, I have revised the manuscript to specify that the measure captured whether participants had the opportunity to take time for themselves on a given day, e.g..”

(Abstract, p. 2): “Results showed that on days when they **had an opportunity for** time to themselves, parents experienced higher positive affect, lower negative affect, and steeper cortisol slopes, indicating better stress recovery”

(Introduction, p. 9): “It was hypothesized that (H1) on days when **individuals have an opportunity for** time to themselves, they report higher positive affect and lower negative affect and exhibit steeper diurnal cortisol slopes ...”

(Results, p. 15): “Participants reported **having had the opportunity** to take personal time on 79% of days (SD = 0.25), averaging about 2 days out of every 8 days without personal time.”

(Results, p. 16): “As hypothesized, parents were more likely to report higher levels of positive affect ($\beta = 0.05$, SE = 0.01, $p < .001$), lower levels of negative affect ($\beta = -0.05$, SE = 0.02, $p = .005$), and steeper diurnal cortisol slopes ($\beta = -0.10$, SE = 0.03, $p = .005$) on days on which they **had the opportunity for** time to themselves.”

(Results, p. 17-18): “Daily associations between **opportunity for** personal time and positive affect were not moderated by personality. The within-person association between **opportunity for** personal time and negative affect was moderated by neuroticism ($\beta = -0.05$, SE = 0.02, $p = .006$) and openness ($\beta = -0.04$, SE = 0.02, $p = .035$).”

(Discussion, p. 20): “Indeed, it was found that parents reported higher levels of positive affect, lower levels of negative affect, and displayed steeper cortisol slopes (indicating better physiological stress recovery) on days on which they **had the opportunity to** take time for themselves, as compared to days without personal time”

(Discussion, p. 20): “Participants reported that they **had the opportunity to** take time for themselves on 4 out of 5 days, on average.”

(Discussion, p. 22): “Specifically, only individuals high (but not those low) in neuroticism showed decreased negative affect and steeper cortisol slopes on days when they had **the opportunity to** take time for themselves.”

(Discussion, p. 24): “The association between **opportunity for** personal time and parents’ daily well-being was not moderated by conscientiousness, agreeableness, or introversion.”

(Conclusion, p. 28): “On days when midlife parents **had the opportunity to** take time for themselves, they reported reduced negative affect and showed steeper cortisol slopes, indicating better stress recovery.”

2. I appreciate your reasons not to have hypotheses about personality- it may be helpful to briefly discuss these reasons in the paper.

Response: As suggested, I have included the reasoning for not having directed hypotheses about how specific personality traits moderate associations of personal time with daily affect and cortisol in the revised paper:

(p. 9): “Directed hypotheses regarding personality traits were not specified because the existing literature on personality differences in the experience of solitude is inconsistent (e.g., with respect to introversion^{48,49,51}), and even less is known about how personality relates specifically to the benefits of personal time.”

3. *It would be helpful to specify the underlying methods used to back up statements of sample differences.. For example “this sample of parents was significantly younger, had higher average household income, better self-rated health, less time to themselves, lower average negative affect, and steeper cortisol slopes than the remainder of the Refresher Daily Diary sample—give some more information to back this up” and “Those excluded due to non-compliance did not differ from included participants in terms of sex, ethnicity, age, subjective health, average personal time, or positive or negative affect, but they did report significantly lower income”.*

Response: Thanks for this comment. The revised manuscript now specifies each test used, as well as the test-statistics and p-values for the sample comparisons:

(p. 11): “Wilcoxon rank sum tests showed that this sample of parents was significantly younger ($W = 119'315$, $p < 0.001$), had higher average household income ($W = 54'798$, $p < 0.001$), better self-rated health ($W = 65'722$, $p = 0.013$), less time to themselves ($W = 93'828$, $p < 0.001$), lower average negative affect ($W = 67'304$, $p = 0.035$), and steeper cortisol slopes ($W = 63'112$, $p < 0.001$) than the remainder of the Refresher Daily Diary sample. Samples did not differ by gender (χ^2 -test; $\chi^2 = 0.19$, $p = .666$).”

(p. 13): “Wilcoxon and χ^2 tests showed that those excluded due to non-compliance did not differ from included participants in terms of sex ($\chi^2 = 0.41$, $p = .524$), ethnicity ($\chi^2 = 1.24$, $p = .266$), age ($W = 2'324$, $p = .562$), income ($W = 2'428$, $p = .325$), subjective health ($W = 1'811$, $p = .342$), average personal time ($W = 1'589$, $p = .094$), or negative affect ($W = 1'986$, $p = .673$), but they did report significantly higher positive affect ($W = 1'422$, $p = .036$).”

4. *Also, it would be helpful to specify your methods for follow-ups to derive estimates for those high/low in neuroticism (I'm assuming simple slopes were tested, specify the process). For example, here Figure 2, individuals high in neuroticism ($\beta = -0.09$, $SE = 0.02$, $p < .001$) and in openness ($\beta = -0.09$, $SE = 0.02$, $p < .001$) experienced a significant decrease in negative affect on days when they reported having had time to themselves. This was not true for individuals low in neuroticism ($\beta = 0.00$, $SE = 0.02$, $p = .937$) or openness ($\beta = -0.02$, $SE = 0.02$, $p = .408$).*

Response: As suggested by the reviewer, I have added a sentence to the statistical analysis section to describe the probing of the interactions:

(p. 15): “Significant interactions were probed by calculating simple slopes at one standard deviation below ($M - 1 SD$) and above ($M + 1 SD$) the mean of the moderator.”